# LOOKAHEAD UNMASKING ELICITS ACCURATE DECODING IN DIFFUSION LANGUAGE MODELS

## ABSTRACT

Masked Diffusion Models (MDMs) as language models generate by iteratively unmasking tokens, yet their performance crucially depends on the inference-time order of unmasking. Prevailing heuristics, such as confidence-based sampling, are myopic: they optimize locally, fail to leverage extra test-time compute, and let early decoding mistakes cascade. We propose **Lookahead Unmasking (LookUM)**, which addresses these concerns by reformulating sampling as *path selection* over all possible unmasking orders without the need for an external reward model. Our framework couples (i) a path generator that proposes paths by sampling from pools of unmasking sets with (ii) a verifier that computes the uncertainty of the proposed paths and performs importance sampling to subsequently select the final paths. Empirically, erroneous unmasking measurably inflates sequence-level uncertainty, and our method exploits this to avoid error-prone trajectories. We validate our framework across six benchmarks, such as mathematics, planning, and coding, and demonstrate consistent performance improvements. LookUM requires only two to three paths to achieve peak performance, demonstrating remarkably efficient path selection. The consistent improvements on both LLaDA and post-trained LLaDA 1.5 are particularly striking: base LLaDA with LookUM rivals the performance of RL-tuned LLaDA 1.5, while LookUM further enhances LLaDA 1.5 itself—showing that uncertainty-based verification provides orthogonal benefits to reinforcement learning and underscoring the versatility of our framework. Code will be publicly released.

## 1 INTRODUCTION

Following the remarkable success of diffusion models in continuous domains such as image and video (Song et al., 2020; Rombach et al., 2022; Ho et al., 2022), diffusion language models (DLMs) have recently emerged as a promising and efficient alternative to autoregressive language models (ARMs) for discrete sequence generation (Austin et al., 2021a; Sahoo et al., 2024; Lou et al., 2023). Notably, state-of-the-art DLMs, including LLaDA (Nie et al., 2025) and Dream (Ye et al., 2025), adopt the form of masked diffusion models (MDMs), where generation proceeds through iterative unmasking over sequences of masked tokens. Recent advances have further enhanced these models through reinforcement learning-based fine-tuning, as demonstrated in LLaDA 1.5 (Zhu et al., 2025) and d1 (Zhao et al., 2025), which employ reward-based optimization to improve reasoning capabilities.

The primary distinction between ARMs and DLMs lies in the order of token unmasking. Unlike ARMs, DLMs possess the flexibility to arbitrarily select the positions of tokens to be unmasked, in addition to predicting token values. For instance, in tasks such as planning (Ye et al., 2024) and infilling (Gong et al., 2024), where bidirectional rather than left-to-right unmasking is expected to be advantageous, DLMs demonstrate clear superiority over ARMs. This highlights that the order of unmasking is not merely an architectural detail but a critical factor in achieving strong performance.

Existing approaches to unmasking predominantly adopt heuristic strategies that rely on token-level model predictions to compute certainty measures (e.g., negative entropy, confidence, margin), which are then used to greedily select the next position to unmask (Chang et al., 2022; Huang et al., 2025; Kim et al., 2025; Koh et al., 2024; Ben-Hamu et al., 2025). Though lightweight and simple, these methods inherently reflect confidence in local token distributions rather than capturing sequence-

Figure 1: **Standard unmasking in discrete diffusion vs. LookUM.** During the denoising process of unmasking from timestep $T$ to 0, greedy approaches often select the position with the highest token-level certainty which can lead to an incorrect unmasking order and result in local errors (**red**). In contrast LookUM generates candidate unmasking paths and leverages a verifier to select those that avoid local errors and recover the correct sequence (**blue**).

level dependencies that govern the overall unmasking trajectory. As a result, they often disregard global consistency and are prone to irreversible local errors. As seen in Figure 1, these errors may resemble small computational mistakes in the unmasking order that propagate forward, which we empirically observe in our experiments in Section 3.1.

These limitations highlight the need for unmasking strategies that can identify potential local errors and steer the unmasking process toward paths that avoid them. To this end, we propose a framework that reformulates sampling as a path selection problem in the space of possible unmasking orders, where a verifier directs the process by detecting local errors and steering it toward more reliable paths. Building on this framework, we develop **Lookahead Unmasking (LookUM)**, which operates in a *fully unsupervised* manner by leveraging a certainty measure over sequences as a verifier.

Our method achieves reliable error avoidance with only a two- to three-fold computational cost. It evaluates candidate paths through a verifier and selects those that best preserve sequence-level coherence. In reasoning benchmarks, LookUM consistently outperforms greedy unmasking, improving performance by up to 4 points on HumanEval and GSM8K, and 8 points on MBPP. Consistent gains in both LLaDA and its RL-tuned variant LLaDA 1.5 demonstrate that our method provides complementary benefits to the model training pipeline.

Overall, our contributions can be summarized as:

- We reformulate the problem of choosing unmasking tokens within a reward guided generation framework that uses verifier scores and sampled paths.

- LookUM reduces local error rates by $10\%$ and improves accuracy by 4-8 points on reasoning benchmarks with only 2-3× overhead.

- The method requires no external models and complements both base training and RL optimization, as shown on LLaDA 1.5.

## 2 BACKGROUND

**Masked Diffusion Language Models.** A widely adopted family of discrete diffusion models for language is the *masked diffusion* framework (Austin et al., 2021a; Lou et al., 2023; Sahoo et al., 2024), which reconstructs a text sequence from fully masked tokens. The process is formulated over timesteps $t \in [0, T]$, comprising (1) a forward noising process that transforms a token sequence $x_0$ into masked tokens, and (2) a reverse denoising process that reconstructs the sequence from the masked token $x_T$. Formally, let $\theta$ denote a DLM operating on sequences of length $L$. At each timestep $t$, the model handles $x_t = (x_t^1, x_t^2, \ldots, x_t^L)$, where each token $x_t^i$ belongs to a vocabulary $\mathcal{V} = \{1, 2, \ldots, m\}$, with the special symbol $m$ indicating the mask token. During denoising, the model sequentially determines the order of token unmasking.

**Decoding process.** The decoding progresses through iterative denoising steps, gradually converting the fully masked sequence $x_T$ into a complete text sequence $x_0$. For each denoising step $t$, let $\mathcal{M}_t := \{i \in \{1, \dots, L\} : x_t^i = m\}$ be a *masked index set* and $p_\theta^i(\cdot \mid x_t)$ be a predicted categorical distribution over $\mathcal{V}$ for each position $i$. Then, the objective of unmasking is to select $b_t$ indices to unmask from the currently masked positions $i \in \mathcal{M}_t$, based on $p_\theta^i(\cdot \mid x_t)$. That is, *unmasking index set $\mathcal{U}_t$* is selected and used to update the next masked index set, such as $\mathcal{M}_{t-1} = \mathcal{M}_t \setminus \mathcal{U}_t$.

In principle, token selection could be random, but the lack of explicit training constraints on the transition dynamics of the denoising process often leads to degraded performance (Ben-Hamu et al., 2025). Therefore, practical implementations employ *greedy* unmasking strategies, which consistently outperform random unmasking. Formally, with a scoring function $\sigma$ over token probability $p_\theta^i$, the unmasking index set is obtained as

$$\mathcal{U}_t = \{i \in \mathcal{M}_t \mid \text{rank}(\sigma_t^i) \leq b_t\}, \tag{1}$$

where $\sigma_t^i = \sigma(p_\theta^i(\cdot|x_t))$ is the score for token probability at position $i$ and $\text{rank}(\cdot)$ is its ranking over $\mathcal{M}_t$. Existing score functions include: (1) **Confidence** (Chang et al., 2022), $\sigma_t^i = p_t^{(i,1)}$; (2) **Margin** (Kim et al., 2025), $\sigma_t^i = p_t^{(i,1)} - p_t^{(i,2)}$; (3) **Negative Entropy** (Koh et al., 2024), $\sigma_t^i = -H(p_\theta^i(\cdot \mid x_t))$ with $H(p_\theta^i) = -\sum_j p_t^{(i,r)} \log p_t^{(i,r)}$, where $p_t^{(i,r)} := p_\theta^i(v_t^{(i,r)} \mid x_t)$ is the probability of the token $v$ with $r$-th highest score over $\mathcal{V}$.

# 3 LOOKAHEAD UNMASKING FOR DIFFUSION LANGUAGE MODELS

This section introduces LookUM for DLMs. We begin by illustrating how local errors arise in greedy unmasking and how sequence-level certainty enables effective path correction. We then reformulate unmasking as a path selection problem guided by certainty-based verifiers, and conclude with the algorithmic design and analysis showing significant performance improvements at low computational cost.

## 3.1 ESCAPING LOCAL ERROR PATHS WITH LOOKAHEAD UNMASKING

As the model tends to unmask tokens only near already unmasked ones (Gong et al., 2025), greedy unmasking often leads to errors that cannot be recovered as local error recovery is impossible (Huang et al., 2025). Unmasking incorrect tokens induces local errors, including mathematical errors (digit misplacement, arithmetic mistakes), coding errors (variable reference errors, missing brackets or syntax), and general errors (illogical reasoning, grammatical inconsistencies). Once such errors occur, the greedy strategy has no mechanism to recover, and the generation becomes locked into an erroneous path.

To address this limitation, we ground our approach in two key observations: (1) local errors propagate by increasing model uncertainty in subsequent predictions, and (2) paths with higher uncertainty exhibit greater susceptibility to local errors. These insights motivate LookUM, which evaluates path-level certainty to selectively avoid error-prone trajectories while preserving sequence-level coherence.

Table 1: **Entropy and Confidence under Local Errors.** Entropy and confidence metrics across arithmetic operations (subtraction, addition, multiplication) demonstrate increased uncertainty when local errors occur versus correct predictions.

| | Correct | | Error | |
|---|---|---|---|---|
| | Entropy↓ | Confidence↑ | Entropy↓ | Confidence↑ |
| Sub. | **0.24** | **0.97** | 1.69 | 0.82 |
| Add. | **0.53** | **0.93** | 1.82 | 0.82 |
| Mul. | **0.30** | **0.96** | 1.61 | 0.83 |

**Local error increase model uncertainty.** We evaluate the impact of local errors on model predictions using the Arithmetic dataset (Brown et al., 2020), which contains 2,000 simple arithmetic problems. By intentionally introducing local errors into the sequences, we compare cases where the model still produces the correct prediction against those affected by errors. We then measure the average entropy and confidence of the generated outputs. As shown in Table 1, the presence of local errors consistently increases prediction uncertainty, reflected by higher entropy values and lower

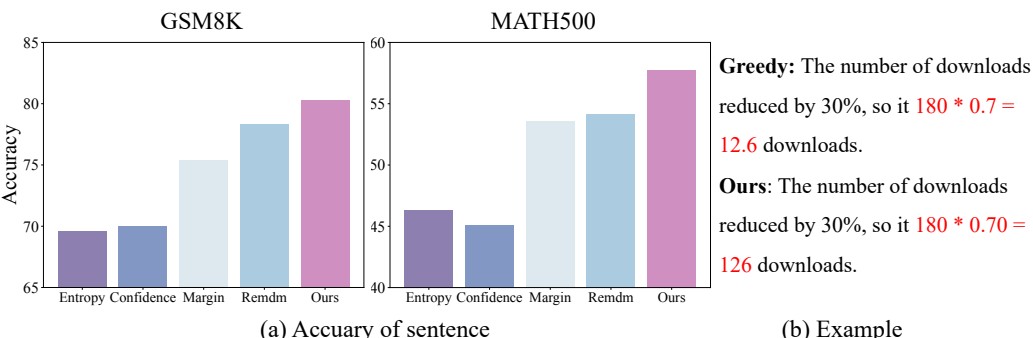

(a) Accuary of sentence          (b) Example

Figure 2: **Local Error Compare and Example.** (a) Sentence-level accuracy on GSM8K and MATH500, showing our method achieves approximately $10\%$ lower error rates than baselines. (b) Example of greedy unmasking producing a computational error ($180 \times 0.7 = 12.6$) while our method generates the correct result (126).

confidence scores. This suggests that local errors propagate instability in the decoding process, ultimately making the model less reliable in its subsequent predictions.

**Avoiding Uncertain Paths.** In the previous analysis, we showed that local errors increase prediction uncertainty and may destabilize subsequent reasoning. Motivated by this observation, we propose Lookahead Unmasking (LookUM), which exploits uncertainty as a signal for guiding the denoising process. The core idea is simple: instead of unmasking tokens in a fixed order, we perform a lookahead search over possible reasoning paths and dynamically adjust unmasking positions according to their certainty.

To demonstrate the effectiveness of this approach, we measure local error rates in MATH500 (Lightman et al., 2023) and GSM8K (Cobbe et al., 2021), using GPT-4o (OpenAI, 2024) to verify the correctness in each reasoning step. As shown in Figure 2, LookUM achieves a consistent reduction of nearly $10\%$ in local error rates compared to baseline methods, confirming that uncertainty-guided path selection effectively avoids error-prone trajectories. The complete algorithmic framework is presented in Section 3.2, with implementation details in Appendix C.

### 3.2 LOOKAHEAD UNMASKING

We reformulate the unmasking procedure in DLMs as a reward guided generation process for selecting unmasking tokens using verifier scores and sampled paths. Our framework consists of two components: a *path generator* that proposes candidate unmasking sets, and an *uncertainty-based verifier* that evaluates their reliability. This design transforms the myopic token-level decisions of existing methods into a lookahead mechanism that considers the downstream consequences of unmasking choices.

At each sampling step $t$, *path generator $G$* leverages the predictive distribution of the model $p_\theta(\cdot \mid x_t)$ together with the available budget $B_t$ to construct a candidate path, that is, an unmasking set $\mathcal{U}_t \subseteq \mathcal{M}_t$. This set specifies which positions are to be unmasked in the current step. The candidate path is then evaluated by the *uncertainty-based verifier $V$*, which, for each position $i \in \mathcal{U}_t$, considers the transformed state where $x_t^i$ is sampled from its token distribution $p_\theta^i(\cdot|x_t)$. By analyzing the one-step-ahead dynamics under this hypothetical unmasking, the verifier assigns an uncertainty score that quantifies the expected consistency of the candidate path with future model predictions. The verifier can be regarded as a reward model that guides the selection of the path. Within this unifying perspective, unmasking can be interpreted as a reward alignment process, allowing its integration with existing methodologies to achieve reward alignment in sampling procedures.

**Path Generator.** Given the state $x_t$, the path generator returns a set of indexes

$$G(p_\theta^i(\cdot|x_t), B_t, \mathcal{P}_t) = \mathcal{U}_t, \quad \mathcal{U}_t \subseteq \mathcal{M}_t, \ |\mathcal{U}_t^{(k)}| = B_t,$$

constructed deterministically or stochastically from a proposal distribution. Deterministic generation can thus be viewed as the conventional top-$B_t$ selection, resulting in a greedy choice of tokens, whereas in the stochastic case, unreliable outcomes may arise if proposals are drawn from low-certainty positions; therefore, the generator must instead sample from a high-certainty pool $\mathcal{P}_t$ with size $|\mathcal{P}_t| = N$, motivating us to propose several alternative constructions of pool as follows:

- **N-best pooling:** construct a candidate pool by selecting the top-$N$ tokens according to a certainty measure. The certainty measure can be defined using standard strategies such as entropy, confidence, or margin, allowing flexible adaptation to different evaluation criteria.
- **Certainty filtering pooling:** include only tokens whose predictive probability exceeds a predefined threshold $\tau$, ensuring proposals are sampled from high-certainty positions.

**Verifier.** For each candidate $\mathcal{U}_t$, we define a look-ahead state $\tilde{x}_{t-1}$, from which the corresponding predictive distribution $p_\theta(\cdot \mid \tilde{x}_{t-1})$ is obtained. The look-ahead state $\tilde{x}_{t-1}$ is determined by sampling from $p_\theta(\cdot \mid x_t)$, following exactly the same procedure as the standard denoising transition. The verifier then evaluates this distribution and assigns an uncertainty score reflecting the expected consistency with future predictions.

$$\tilde{x}_{t-1}^i \sim p_\theta^i(\cdot \mid x_t), \ \ \forall i \in \mathcal{U}_t, \qquad V(p_\theta(\cdot|\tilde{x}_{t-1})) \in \mathbb{R}.$$

We propose to quantify uncertainty by leveraging the probability distribution generated by the model and employ this measure as the basis of the verifier. The verifier operates in intermediate sampling states, but evaluates them using a sequence-level approximation to the prediction of the model of $x_0$. The verifier operates in intermediate sampling states, quantifying the potential certainty over the entire sequence of tokens. We next introduce several candidate functions that can serve as verifiers:

- **Average Negative Entropy:** Instantiate the verifier with the mean Shannon entropy of the predictive distribution induced by the model under the look-ahead state. To maintain consistency with the other verifiers and to interpret entropy as a certainty measure, we adopt the negative entropy, so that larger scores correspond to higher certainty. For each position $i \in [L]$, define

$$V(p_\theta(\cdot|\tilde{x}_{t-1})) = -\frac{1}{L} \sum_{i \in [L]} H(p_\theta^i(\cdot|\tilde{x}_{t-1})).$$

  Entropy aggregates information across the entire sequence distribution. This makes it sensitive not only to the dominance of the leading token but also to the dispersion of probability mass among lower-ranked alternatives.

- **Average Confidence:** Instantiate the verifier with the mean of the maximum probabilities of the predictive distributions under the look-ahead state. For each position $i \in [L]$, define

$$V(p_\theta(\cdot|\tilde{x}_{t-1})) = -\frac{1}{L} \sum_{i \in [L]} \max_{v \in \mathcal{V}} \ p_\theta^i(v|\tilde{x}_{t-1}).$$

  Confidence reflects the degree to which the model concentrates probability mass on a single outcome, thereby serving as a straightforward indicator of positional certainty across the sequence.

The interaction between the path generator and the verifier can be naturally viewed as a two-stage sampling process: the generator proposes candidate unmasking paths, and the verifier scores these candidates based on predictive consistency. By preferentially selecting paths with lower uncertainty, this process aligns generation with more reliable trajectories, similar to how reward models guide sampling in reinforcement learning. From this perspective, the unmasking can be interpreted as a form of reward alignment, where the verifier acts as a surrogate reward model. The verifier evaluates the entire sequence, which corresponds to the expected value of future rewards.

We adopt two sampling schemes: Sequential Monte Carlo (SMC), which propagates weighted particles through the denoising process with incremental reweighting, and Nested Importance Sampling (NIS), which performs importance weighting at each step based on immediate reward estimates. These approaches follow established methodologies for reward-aligned sampling in previous work (Li et al., 2024; Wu et al., 2023). The concrete implementation details of both methods are provided in Appendix B.

### 3.3 Algorithm Design and Analysis

The proposed LookUM combines path generation with uncertainty verification in a unified framework, thereby guiding the model away from erroneous unmasking paths. The design of the algorithm depends on the specific choice of verifier, path generator, and sampling scheme, while its computational complexity is determined by the number of candidate paths generated during each iteration. This section provides a formal design of the algorithm and analyzes its computational properties.

**Algorithm.** Algorithm 1 illustrates the overall pseudocode procedure. Starting from a fully masked sequence $x_T$, the algorithm iteratively reduces the mask set $\mathcal{M}t$ until a complete sequence $x_0$ is obtained. At each step $t$, the path generator $G$ produces $k$ candidate unmasking sets $\{\mathcal{U}_{t,i}\}_{i=1}^{k}$, which are sampled from a certainty-based pool $\mathcal{P}_t$ (lines 4-5). Each candidate defines a hypothetical next state $\tilde{x}_{t-1,i}$ by unmasking tokens at the selected positions according to their predictive distributions. The verifier $V$ then assigns uncertainty-based scores $\{s_i\}_{i=1}^{k}$ to these states (line 6), and the actual next state $\tilde{x}_{t-1}$ follows from the selection procedure detailed in Appendix B (line 7).

---

**Algorithm 1** Algorithm of LookUM

1: **Require:** Verifier $V$, path generator $G$, diffusion model $p_\theta$, budget for each timesteps $\{B_t\}_{t=T}^{1}$, lookahead number k.
2: **Init:** Masked sequence $x_T$
3: **for** $t \in \{T, \cdots, 1\}$ **do**
4:      Model prediction with $p_\theta(\cdot|x_t)$ and make pool $\mathcal{P}_t$
5:      generate $k$ paths $\{\mathcal{U}_{t,i}\}_{i=1}^{k}$ with generator $G(p_\theta(\cdot|x_t), B_t, \mathcal{P}_t)$ and sample candidates $\{\tilde{x}_{t-1,i}\}_{i=1}^{k}$ using $\{\mathcal{U}_{t,i}\}_{i=1}^{k}$         ▷ Path Generation
6:      Evaluate future predictions $\{p_\theta(\cdot|\tilde{x}_{t-1,i})\}_{i=1}^{k}$ using verifier $V$ to obtain uncertainty scores $\{s_i\}_{i=1}^{k}$         ▷ Verification
7:      Select $\tilde{x}_{t-1}$ from $\{\tilde{x}_{t-1,i}\}_{i=1}^{k}$ using importance sampling (SMC/NIS) based on scores $\{s_i\}_{i=1}^{k}$         ▷ Selection
8: **end for**

---

**Complexity Analysis.** The additional computational overhead of Lookahead Unmasking arises primarily from the evaluation of $k$ paths at each step. The construction of the pool $\mathcal{P}_t$ and the operations of the verifier $V$ introduce only minimal computational overhead, as they are based on tensor computations on already predicted distributions and remain negligible compared to the cost of model inference. As a result, LookUM incurs $k$ times the computational cost of the standard decoding procedure. We analyze the performance variation with respect to the number of generated paths in Section 4.3, and the results are presented in Figure 3. The scaling effect converges rapidly at low computational cost, and using 2 to 3 paths, which corresponds to the cost of commonly employed classifier free guidance method (Ho & Salimans, 2022), is sufficient to achieve near-optimal performance. In addition, a detailed analysis of wall-clock time is provided in Appendix D.3.

## 4 Experiments

### 4.1 Experimental Setup

**Datasets.** We evaluate LookUM on diverse reasoning benchmarks, including MATH500 (Lightman et al., 2023) and GSM8K (Cobbe et al., 2021) for mathematics, HumanEval (Chen et al., 2021) and MBPP (Austin et al., 2021b) for coding, and Sudoku and Countdown for planning tasks.

**Baselines.** We compare LookUM with *five* unmasking strategies: (1) Confidence-based unmasking (Chang et al., 2022) that sequentially unmasks tokens in descending order of prediction confidence, (2) Margin-based unmasking (Kim et al., 2025) that unmasks tokens based on the margin between their top-1 and top-2 prediction probabilities, (3) Entropy-based unmasking (Koh et al., 2024) that unmasks tokens in ascending order of entropy, and (4) PC-Sampler (Huang et al., 2025) that calibrates the original confidence by incorporating position-aware weighting and frequency-based adjustment. In addition, although they are not Unmasking Strategies, we also experiment

with (5) ReMDM (Wang et al., 2025) which alter model predictions by incurring additional computational cost. For a fair comparison, we also report results in which all baselines, except ReMDM which already operates at a comparable computational cost to LookUM, use 3 paths followed by majority voting.

**Implementation Details.** We use LLaDA-8B-Instruct (Nie et al., 2025) and LLaDA-1.5 (Zhu et al., 2025) for main experiments. Following the decoding setup in (Zhao et al., 2025), we set the sequence lengths to 128 and 256 and apply unmasking with two tokens per step. All experiments are conducted using two NVIDIA A100 GPUs.

## 4.2 MAIN RESULTS

Table 2 demonstrates that Lookahead Unmasking consistently outperforms baseline sampling strategies across both LLaDA-8B and the RL-tuned LLaDA 1.5, without requiring additional model fine-tuning. The method achieves substantial improvements: an absolute gain of 8 points on HumanEval at sequence length 128 and 4 points on GSM8K for base LLaDA. Notably, on LLaDA 1.5, which has already undergone reinforcement learning optimization, LookUM still delivers significant gains. This demonstrates that our uncertainty-based approach captures complementary signals to reward-based training, as it can further enhance models that have already been optimized through RL.

The consistent improvements across both base and RL-tuned models provide strong empirical evidence that Lookahead Unmasking effectively avoids erroneous unmasking paths. Particularly striking is that LLaDA-8B with LookUM achieves competitive or superior performance to vanilla LLaDA 1.5 on several benchmarks, suggesting that inference-time optimization can partially substitute for expensive RL training.

While ReMDM attempts to mitigate local errors by remasking low-confidence tokens, its performance gains are inconsistent. Results on Countdown, HumanEval, and MBPP show minimal improvement or degradation, indicating that remasking may disrupt solution trajectories in tasks requiring global structural coherence. By preserving the natural generative path without altering established structure, Lookahead Unmasking delivers more reliable improvements across diverse benchmarks.

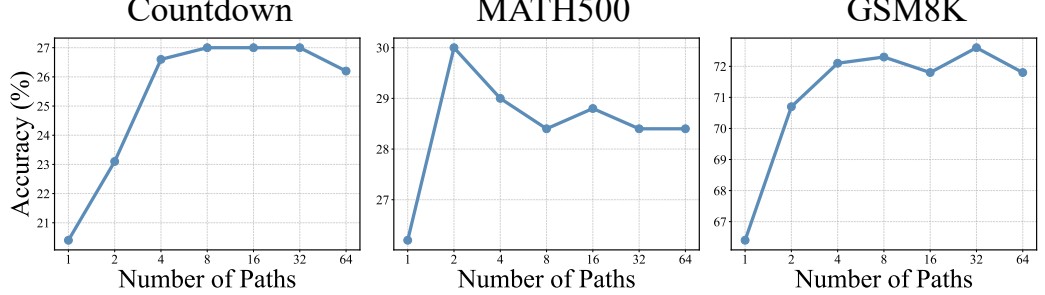

Figure 3: **Figure of scaling results.** Performance scaling with lookahead paths. Accuracy versus number of particle paths on three benchmarks (GSM8K, MATH500, Countdown). Sharp improvements occur up to 4 particles, after which performance saturates, demonstrating efficient scaling with limited computational budget.

## 4.3 ANALYSIS

**Experiments with External Reward Model.** A key advantage of LookUM is its independence from external reward models, eliminating the need for costly task-specific training or model deployment. To validate this design choice, we investigate whether incorporating mathematical process reward models could enhance performance. We employ Qwen2.5-Math-PRM-7B (Zhang et al., 2025) as an alternative verifier and evaluate it using both NIS and SMC sampling methods with varying numbers of particles (1, 4, 16) on MATH500 and GSM8K.

Table 2: **Performance comparison across models.** Results on six reasoning benchmarks with LLaDA-8B and LLaDA-1.5 (RL-tuned). Best results per model in **bold**, second-best underlined.

| Model | Method | MBPP | | Humaneval | | GSM8K | | MATH500 | | Countdown | | Sudoku | |
|---|---|---|---|---|---|---|---|---|---|---|---|---|---|
| | | 128 | 256 | 128 | 256 | 128 | 256 | 128 | 256 | 128 | 256 | 128 | 256 |
| LLaDA | Confidence (× 1) | 28.6 | 28.4 | 19.5 | 32.0 | 68.3 | 76.7 | 26.0 | 32.4 | 20.3 | 21.9 | 1.4 | 27.4 |
| | Confidence (× 3) | - | - | - | - | 68.8 | 77.7 | 22.8 | 28.8 | 19.5 | 12.1 | - | - |
| | Margin (× 1) | 28.6 | 28.4 | 25.6 | 31.0 | 67.1 | 76.1 | 28.4 | 34.4 | 19.1 | 20.7 | 21.8 | 27.8 |
| | Margin (× 3) | - | - | - | - | 70.6 | 77.1 | 23.4 | 30.2 | 17.3 | 11.3 | - | - |
| | Entropy (× 1) | 27.2 | 24.4 | 19.5 | 15.9 | 66.7 | 75.4 | 26.0 | 33.0 | 21.9 | 20.3 | 0.0 | 12.0 |
| | Entropy (× 3) | - | - | - | - | 59.6 | 69.9 | 22.2 | 27.8 | 13.4 | 10.2 | - | - |
| | PC-Sampler (× 1) | 24.0 | 25.2 | 13.4 | 30.5 | 67.3 | 73.7 | 25.2 | 32.4 | 26.5 | 20.3 | 23.2 | 24.0 |
| | PC-Sampler (× 3) | - | - | - | - | 66.0 | 75.0 | 21.8 | 27.6 | 14.5 | 12.0 | - | - |
| | ReMDM | 28.6 | 28.4 | 17.1 | 29.3 | 69.1 | 77.9 | 27.4 | 33.0 | 25.3 | 17.2 | 0.4 | 22.8 |
| | **LookUM** | **30.5** | **36.2** | **27.4** | **35.9** | **72.7** | **79.3** | **28.8** | **34.6** | 25.4 | **23.1** | **25.0** | **28.0** |
| LLaDA-1.5 | Confidence (× 1) | 40.3 | 38.6 | 26.8 | 23.7 | 69.5 | 79.4 | 28.6 | 32.6 | 20.3 | 23.4 | 1.4 | 27.4 |
| | Confidence (× 3) | - | - | - | - | 71.1 | 79.7 | 27.3 | 29.2 | 20.1 | **23.5** | - | - |
| | Margin (× 1) | 39.5 | 38.1 | **31.1** | 27.4 | 71.3 | 78.3 | 27.2 | 35.0 | 24.6 | 14.0 | 21.8 | 27.8 |
| | Margin (× 3) | - | - | - | - | 71.4 | 78.4 | 24.4 | 30.6 | 24.6 | 14.1 | - | - |
| | Entropy (× 1) | 40.7 | 36.5 | 20.7 | 23.2 | 69.7 | 77.0 | 28.2 | 32.2 | 23.0 | 12.9 | 0.0 | 12.0 |
| | Entropy (× 3) | - | - | - | - | 71.1 | 80.1 | 22.6 | 29.8 | 15.1 | 12.7 | - | - |
| | PC-Sampler (× 1) | 42.8 | 39.6 | 23.8 | 23.8 | 70.1 | 77.3 | 26.6 | 32.2 | 25.4 | 19.1 | 25.6 | 27.2 |
| | PC-Sampler (× 3) | - | - | - | - | 69.9 | 79.5 | 22.6 | 34.4 | 19.8 | 14.5 | - | - |
| | ReMDM | 41.9 | 39.3 | 28.1 | 29.8 | 70.4 | 80.1 | 27.4 | 34.0 | 23.4 | 19.9 | 0.4 | 22.8 |
| | **LookUM** | **45.0** | **43.6** | 30.7 | **33.5** | **74.5** | **82.3** | **29.2** | **35.8** | **27.3** | 17.9 | **26.8** | **28.0** |

Table 3 shows reward-based verification underperforms our uncertainty-based approach. Reward models expect coherent partial solutions but encounter noisy (Gong et al., 2025), globally-distributed predictions in intermediate states of DLMs. This incompatibility validates our use of intrinsic uncertainty signals, confirming that model-free design of LookUM is more effective and practical for diffusion language models.

**Inference Time Scaling.** The SMC-based process LookUM is extensible with respect to the number of particles used. We regard the number of particles as the scaling axis and investigate the inference-time scaling effect. We use the *number of paths* generated by the path generator as our measure of computational cost, as each path requires a separate model evaluation while the computational overhead of verifier remains negligible. The experiments are conducted with LLaDA on GSM8K, MATH500, and Countdown, with the generation length fixed at 128 and the number of paths set to $k \in \{1, 2, \ldots, 64\}$.

Table 3: **Performance with external reward models**

| PRM | MATH500 | | | GSM8K | | |
|---|---|---|---|---|---|---|
| | 1 | 4 | 16 | 1 | 4 | 16 |
| NIS | 26.0 | 26.6 | 26.0 | 69.0 | 69.1 | 69.5 |
| SMC | 26.0 | 22.6 | 24.0 | 69.0 | 68.0 | 68.5 |

As illustrated in Figure 3, we observe distinct scaling patterns across benchmarks. GSM8K shows substantial improvements up to 4 paths, after which performance plateaus. MATH500 achieves optimal performance with just 2 paths, with additional paths leading to slight performance degradation. These results reveal that LookUM reaches optimal performance with merely 2-4 paths, requiring comparable computational overhead to classifier-free guidance. Rapid saturation indicates that our uncertainty-based verifier efficiently identifies promising unmasking trajectories without exhaustive search, making the method both practical and robust for deployment under computational constraints.

**Multipath Decoding Fails for Baselines** When evaluating the baseline models using multiple paths, we observed that the results were largely comparable to the single-path setting and, in some cases, even worse. This indicates that simply increasing the number of paths is not an effective strategy for discrete diffusion models. In tasks such as MATH and Countdown, where the model exhibits a relatively high error rate, generating multiple paths often led to the model repeatedly selecting the same incorrect trajectory, rather than providing additional diversity. These findings show that, unlike naive path expansion, LookUM offers not only effective performance improvements but also a computationally efficient scaling mechanism.

**Partial Application of Lookahead.** While LookUM applies lookahead refinement at every denoising step, it is natural to ask whether applying lookahead only at a subset of timesteps may retain most of the performance benefits while reducing inference cost. To investigate this question, we divide the generation process into four equal stages and apply lookahead unmasking exclusively within each stage. This setup allows us to systematically evaluate the contribution of lookahead refinement at different portions of the unmasking trajectory.

As shown in Table 4, applying lookahead only within a single stage provides modest improvements compared to the base sampler but remains consistently below the performance of the full LookUM method. Interestingly, no single interval emerges as dominant across tasks; LookUM's advantage appears to arise from the cumulative corrections performed across the entire trajectory. This suggests that local refinement is not confined to a particular region of the unmasking process; instead, errors can occur at different stages, and LookUM must remain active throughout denoising to fully mitigate them.

Table 4: **Performance when LookUM is applied within specific stages.**

| Dataset | ∼0.25T | 0.25T–0.5T | 0.5T–0.75T | 0.75T–T |
|---|---|---|---|---|
| GSM8K | 69.8 | 70.0 | 70.2 | 69.7 |
| MATH500 | 27.8 | 27.0 | 27.8 | 27.9 |
| Countdown | 25.0 | 22.7 | 20.1 | 21.5 |

### 4.4 Component Discovery

Table 5: **Ablation study** of LookUM components on GSM8K, MATH500 and Countdown.

| Component | Variant | GSM8K | MATH500 | Countdown |
|---|---|---|---|---|
| Path Generator | N-best (Confidence) | 72.1 | **29.0** | **32.0** |
| | N-best (Margin) | 72.3 | 28.6 | 31.3 |
| | N-best (Negative Entropy) | 72.5 | 28.8 | 28.1 |
| | Certainty Filtering | **72.6** | 26.4 | 19.5 |
| Verifier | Avg. Negative Entropy | **72.3** | **28.6** | **31.3** |
| | Avg. Confidence | 71.0 | 27.8 | 30.5 |
| Sampling | SMC | 70.7 | **29.0** | 23.1 |
| | NIS | **72.3** | 28.6 | **31.3** |

We conduct ablation studies to analyze the contribution of each component in LookUM: the path generator, verifier, and sampling method. Table 5 presents results on GSM8K, MATH500, and Countdown with sequence length 128 and 64 denoising steps, varying one component at a time while keeping others fixed. Full experimental details are provided in Appendix C.

**Path Generator.** We evaluate four pooling strategies: N-best with confidence, margin, and entropy criteria, and certainty filtering. Results show that N-best pooling variants achieve comparable performance on GSM8K, while certainty filtering yields the highest accuracy on GSM8K but degrades significantly on MATH500 and Countdown. This suggests that adaptive threshold-based selection is less robust across different task complexities than rank-based selection.

**Verifier.** Average negative entropy outperforms average confidence across all benchmarks, with particularly notable improvements on MATH500. The entropy-based verifier better captures distributional uncertainty across the entire sequence, whereas confidence focuses only on the maximum probability, potentially missing important uncertainty signals in lower-ranked tokens.

**Sampling Method.** NIS and SMC show complementary strengths across different tasks. While SMC achieves better performance on MATH500, NIS demonstrates superior results on GSM8K and particularly striking improvements on Countdown.

## 5 RELATED WORK

**Unmasking Strategies for Masked Diffusion Models.** MDMs reconstruct sequences through iterative predictions during the Unmasking process. The order-agnostic training of MDMs permits diverse unmasking paths during sampling, and a variety of strategies have been proposed to exploit this flexibility for performance improvement. A baseline approach selects tokens for unmasking at random, consistent with the training procedure, but this strategy often leads to performance degradation in specific tasks. The High-Confidence Unmasking strategy (Chang et al., 2022) unmask tokens with the highest confidence in the intermediate prediction of the model, and it has been employed in LLaDA. The Top Probability Margin strategy (Kim et al., 2025) selects tokens for unmasking based on the difference between the two highest probability values at each position, and it has demonstrated substantial performance gains in logic puzzles. Dream further introduces a strategy that unmask tokens at positions with low entropy.

P2 (Peng et al., 2025) addresses a core limitation of standard masked diffusion decoding, where tokens cannot be revised once they have been unmasked. To overcome this constraint, P2 proposes a planning-based mechanism that identifies positions to remask and reevaluate during inference. The method formulates the remasking problem from a path-planning perspective and extends the ELBO objective to determine which locations should be remasked. However, while P2 focuses on where to remask, LookUM focuses on where to unmask next in order to make more reliable forward decoding decisions. From this perspective, the two approaches are complementary.

**Reward Guided Path Optimization.** Reward-based path optimization has been explored in several studies, including works on protein design that employ approaches such as SMC (Doucet et al., 2001), SVDD (Li et al., 2024) and FK Steering (Singhal et al., 2025). However, task-specific reward models often need to be trained separately, which not only limits scalability but also makes their application to complex tasks such as mathematics and code generation particularly challenging. In contrast, our method assesses the model's predictions via an intrinsic uncertainty-oriented function, thereby obviating the reliance on an external reward model. In contrast, our method assesses predictive entropy, thereby obviating the reliance on an external reward model.

## 6 CONCLUSION

We proposed Lookahead Unmasking, an unsupervised inference-time framework that reformulates unmasking as a path selection problem guided by uncertainty. Lookahead Unmasking effectively reduces local errors, scales with additional compute, and consistently improves reasoning performance across mathematics, coding, and planning benchmarks, without relying on external reward models. Our results demonstrate that uncertainty-aware unmasking offers a simple and general approach for advancing diffusion language models.

Future work might fruitfully investigate verifiers that examine intermediate model representations beyond output probabilities, particularly attention patterns across layers and timesteps. Such exploration could potentially reveal which token relationships the model prioritizes during unmasking, possibly offering signals orthogonal to uncertainty for path selection. We speculate that combining multiple intrinsic signals (uncertainty, attention weights, gradient magnitudes) could lead to more robust verifiers that may better capture the internal reasoning process of the model. Exploring these internal dynamics might open new avenues for sophisticated path selection strategies that could extend beyond surface level output analysis.

## REPRODUCIBILITY STATEMENT

We provide hyperparameter details and setup of all experiments in Section 4.1 and Appendix C.

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

## A  LIMITATIONS

While our work demonstrates that uncertainty provides a strong signal for path selection, other intrinsic model signals might offer complementary benefits. Our LookUM suggests that uncertainty-based verifiers may be particularly well suited for evaluating the noisy intermediate states characteristic of diffusion models, though this approach relies solely on model output distributions. In continuous diffusion domains, attention manipulation has shown promise for guiding generation (Ahn et al., 2024), yet the potential for leveraging such internal representations in discrete diffusion remains largely unexplored.

## B  SAMPLING ALGORITHM FOR LOOKAHEAD UNMASKING

### B.1  IMPORTANCE SAMPLING

Importance Samplings are widely employed to approximate expectations of complex functions with respect to a probability distribution. Specifically, given a target distribution $p^*(x)$, the goal is to compute

$$\mu = \mathbb{E}_{p^*}[f(x)] = \int f(x)\, p(x)\, dx. \tag{2}$$

In many cases direct sampling from $p^*(x)$ is difficult, or rare events of interest occur with extremely low probability. Which alternative proposal distribution $p(x)$, from which sampling is easier, and reweight the samples accordingly:

$$\mu = \int f(x)\, p(x)\, dx = \int f(x)\, \frac{p^*(x)}{p(x)}\, p(x)\, dx = \mathbb{E}_p[f(x)\, w(x)], \tag{3}$$

where $w(x) = \frac{p(x)}{q(x)}$ is referred to as the *importance weight*. By choosing $f$ as a Dirac-delta function, the procedure can be viewed as an instance of important resampling. In this setting, normalized weights enable sampling that approximates draws from $p^*(x)$. The pseudocode of Important Sampling is presented as follows:

---

**Algorithm 2** Importance Sampling

---

**Require:** Target distribution $p^*(x)$, proposal distribution $p(x)$, function $f(x)$, number of samples $N$
  1: **for** $i = 1$ to $N$ **do**
  2:    Sample $x_i \sim p(x)$
  3:    Compute weight $w_i \leftarrow \frac{p^*(x_i)}{p(x_i)}$
  4: **end for**
  5: Normalize weights: $\tilde{w}_i \leftarrow \dfrac{w_i}{\sum_{j=1}^{N} w_j}$
  6: Sample $x \sim \sum_{i=1}^{N} \tilde{w}_i\, \delta_{x_i}$

---

### B.2  SEQUENTIAL MONTE CARLO

Sequential Monte Carlo (SMC), often referred to as particle filtering, is a family of algorithms for approximating sequences of probability distributions like diffusion models. The idea is to propagate a set of weighted samples through time using importance sampling and resampling steps. The algorithm SMC is as follows:

---

**Algorithm 3** Sequential Monte Carlo

---

1: **Require:** number of particles $N$, initial distribution $p(x_0)$, proposal kernels $p(\cdot \mid x_{t-1})$, target kernels $p^*(\cdot \mid x_{t-1})$
2: **Initialization:**
3: **for** $i = 1, \ldots, N$ **do**
4:     Draw $N$ samples $x_0^{(i)} \sim p(x_0)$
5:     Set weight $w_0^{(i)} = 1$
6: **end for**
7: **for** $t = 1, \ldots, T$ **do**
8:     **Propagation:** Draw $N$ samples $x_t^{(i)} \sim q_t(\cdot \mid x_{t-1}^{(i)})$
9:     **Weighting:**
$$w_t^{(i)} = \frac{p_t^*(x_t^{(i)})}{p_{t-1}^*(x_{t-1}^{(i)}) \, p_t(x_t^{(i)} \mid x_{t-1}^{(i)})}$$
10:     Normalize $\tilde{w}_t^{(i)} = w_t^{(i)} / \sum_{j=1}^{N} w_t^{(j)}$
11:     **Resample:** Replace $\{x_t^{(i)}\}_{i=1}^{N}$ with indices drawn according to $\tilde{w}_t^{(1:N)}$.
12: **end for**
13: **Output:** weighted particle system $\{x_{0:T}^{(i)}, \tilde{w}_T^{(i)}\}_{i=1}^{N}$

---

### B.3 Reward Guided Sampling

The objective of reward-aligned sampling is to maintain the naturalness of generated samples while optimizing for the specified reward, or more broadly, to generate samples from a target distribution that encodes the underlying preferences. Concretely, the objective of reward-guided sampling with a reward function $r(\cdot)$ is to draw samples from the target distribution:

$$p^*(\cdot) = \arg\max_p \mathbb{E}_{x_0 \sim p(\cdot)}\big[r(x_0)\big] - \alpha \, D_{\mathrm{KL}}\left(p(x_0) \| p(x_0)\right) \propto \exp\big(r(\cdot)/\alpha\big)p_\theta(\cdot)$$

where $\alpha$ controls the strength of the KL divergence regularization term. The reward is a measure of how well the model output matches the desired criteria. It can be broadly defined to include classifier evaluations, correctness, and other factors.

A diffusion model generates samples sequentially and each transition kernel must be optimized to enable sampling from the target distribution. The optimal transition kernel can be derived as follows:

$$p^*(x_{t-1}|x_t) \propto p_\theta(x_{t-1}|x_t)\exp(r(x_{t-1})/\alpha), \text{where} \quad r(x_{t-1}) = \alpha \log \mathbb{E}_{x_0 \sim p_\theta(\cdot|x_t)}\left[\exp\left(\frac{r(x_0)}{\alpha}\right)\right].$$

Intermidiate reward $r(x_{t-1})$ determines how strongly the model should steer the denoising trajectory toward reward-aligned regions of the sample space.

However, in discrete diffusion $x_0$ is not directly available during sampling. Following standard practice in diffusion-based optimal control, we use a log–exp first-order approximation:

$$r(x_t) \approx \mathbb{E}_{x_0 \sim p_\theta(\cdot|x_t)}\left[r(x_0)\right].$$

This approximation yields a tractable surrogate reward that can be computed from the model's predictive distribution at $x_t$.

Under this formulation, the verifier's uncertainty-based score becomes an explicit approximation to the $r(x_t)$. Different verifier choices correspond to different surrogate rewards:

- **Negative entropy score.**
$$\mathbb{E}[r(x_0)] = \mathbb{E}_{x_0 \sim p_\theta(\cdot|x_t)}\left[\log p_\theta(x_0 \mid x_t)\right] = -H(p_\theta(x_0 \mid x_t)),$$
  which evaluates the overall uncertainty of the predictive distribution at $x_t$.
- **Confidence score.**
$$\mathbb{E}[r(x_0)] \approx \max_{x_0} p_\theta(x_0 \mid x_t),$$
  which assigns higher reward to states with more concentrated model predictions.

Thus, the verifier provides a principled surrogate reward that evaluates the desirability of each intermediate state in a manner consistent with the KL-regularized objective.

Existing approaches to approximate the optimal transition kernel using Sequential Monte Carlo (Singhal et al., 2025) or Nested Importance Sampling (Li et al., 2024). Nested Importance Sampling(NIS) is a method that estimates expectations by applying importance sampling at every stage of a sequential process. The weight $w_t^{(i)}$ for each sampling step is given by:

$$\text{NIS: } w_t^{(i)} = \exp\left(r(x_t^{(i)})/\alpha\right) \quad \text{SMC: } w_t^{(i)} = \exp((r(x_{t-1}) - r(x_t))/\alpha),$$

where the generative process is defined in reverse time, in a manner that is compatible with the denoising process.

## C  EXPERIMENTAL DETAILS

### C.1  EXPERIMENTAL DETAILS IN SECTION 3.1

**Table 1.**   We use arithmetic datasets (subtraction, addition, multiplication) with 2,000 two-number calculations each. Generation length is set to 8 tokens, with arithmetic questions serving as prompts. To analyze local error impact, we use GPT-4o to create erroneous versions by introducing computational mistakes, then compare mean negative entropy and mean confidence between correct and error-containing trajectories. This controlled setup quantifies how local errors affect model uncertainty during generation.

**Figure 2.**   The generation was performed with the same setup as the main experiment, using LLaDA-8B-Instruct with a sequence length of 128 and 64 steps, and a temperature of 0.1. For Lookahead Sampling, we set $|\mathcal{P}_t| = 5$, selected two paths, and verify samples using Average Negative Entropy. We measure the error rate by counting local errors at the sentence level using GPT-4o. The system prompt is configured to strictly evaluate only local errors at the sentence level. The experimental results are presented in Figure 2.

### C.2  EXPERIMENTAL DETAILS IN SECTION 4

**Evaluation Hyperparameters.**   In the experiments, we evaluated two types of DLMs: LLaDA, Dream, and LLaDA-1.5. For all two experiments, Lookahead Sampling was performed with $|\mathcal{P}_t| = 5$, two selected paths, and the Average Negative Entropy setting. The sequence lengths were set to 128 and 256, and unmasking with two tokens per step was applied in all cases. The baselines, ReMDM and PC-Sampler, were implemented following the original implementations provided in their respective papers.

**With External Reward Model.**   We consider Qwen2.5-Math-PRM-7B as the reward model. The experiments are conducted with LLaDA, and at each step we apply the Resample stage of SMC and NIS. At every step, we evaluate the prediction of $x_0$ and adopt $\alpha = 0.1$, Confidence Unmasking which is commonly used in prior work.

**Ablation**   We conduct an ablation study by varying the path generator, verifier, and sampling method while keeping the evaluation setting as the default. All experiments are performed with a sequence length of 128, temperature 0.2, $|\mathcal{P}_t| = 5$, $\tau = 0.6$ and the results are compared on MATH500,GSM8K and Countdown benchmarks.

## D  ADDITIONAL EXPERIMENTS

### D.1  ABLATION OF CERTAINTY FILTERING THRESHOLD

To find the optimal Certainty Filtering threshold, we conducted a ablation over threshold. Specifically, we varied the threshold from 0.9 down to 0.1 in increments of 0.1 and evaluated performance on all language generation tasks. Table C.2 reports the full results.

Table 6: **Ablation of Certainty Filtering Threshold.** Performance across thresholds from 0.9 to 0.1.

| Threshold | 0.9 | 0.8 | 0.7 | 0.6 | 0.5 | 0.4 | 0.3 | 0.2 | 0.1 |
|---|---|---|---|---|---|---|---|---|---|
| GSM8K | 68.3 | 69.1 | 70.1 | 72.6 | 70.1 | 70.1 | 69.2 | 67.5 | 67.5 |
| MATH500 | 27.4 | 27.6 | 26.4 | 26.4 | 26.1 | 26.2 | 26.2 | 24.4 | 23.1 |
| Countdown | 20.7 | 20.7 | 20.5 | 19.5 | 21.1 | 23.1 | 23.1 | 20.3 | 20.1 |

### D.2 CORRELATION OF UNCERTAINTY AND ACCURACY

To examine whether the verifier score is quantitatively correlated with reasoning accuracy, we conducted an additional analysis. Specifically, for each problem in the MATH dataset, we generated 10 reasoning paths and accumulated the verifier scores selected during the generation process at each timestep to compute a final aggregated score. Figure 4 visualizes the relationship between this aggregated score and correctness.

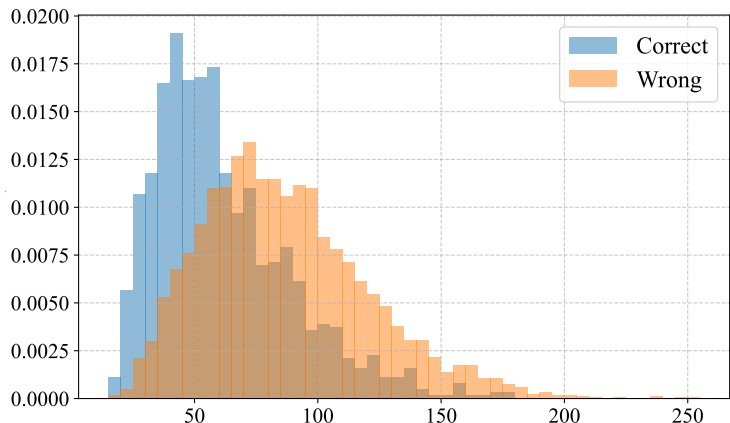

Figure 4: **Histogram of accuracy via entropy**

To verify that this trend reflects a statistically significant difference rather than random variation, we compared the score distributions of the correct and incorrect groups using several nonparametric statistical measures. Cliff's delta (Cliff, 1993), which quantifies the effect size between the two distributions, was 0.4930, indicating a large degree of separation.

In addition, the Kolmogorov–Smirnov test (Smirnov, 1948), which measures the maximum discrepancy between the empirical cumulative distribution functions, KS statistic of 0.3610, with $p < 1.77 \times 10^{-108}$, demonstrating extremely strong statistical significance. Mann–Whitney U test (Mann & Whitney, 1947) likewise indicated a substantial difference with $p < 2.70 \times 10^{-130}$.

Result indicate that the verifier score provides a reliable signal that consistently separates correct from incorrect reasoning trajectories.

### D.3 WALL-CLOCK TIME RESULTS

experiments were conducted with LLaDA, and after a brief GPU warm-up, we computed the mean and standard deviation over 20 generations for both the baselines and LookUM. The results are summarized in Table 7.

The measurements show that LookUM achieves substantially lower wall-clock time than its nominal computational cost would suggest, owing to its parallelizable operations. In contrast, ReMDM exhibits a significant increase in runtime due to its sequential remasking steps. Moreover, LookUM attains its full effectiveness with as few as three paths, enabling it to benefit from parallelism without incurring noticeable performance degradation.

Table 7: Wall-clock time comparison across methods.

| Setting | LookUM | ReMDM | Entropy | Confidence | Margin | PC-sampler |
|---|---|---|---|---|---|---|
| **1 path** | – | – | $4.34 \pm 0.21$ | $4.30 \pm 0.17$ | $4.53 \pm 0.37$ | $4.27 \pm 0.15$ |
| **3 paths** | $7.34 \pm 0.42$ | $16.70 \pm 0.21$ | $7.07 \pm 0.49$ | $6.42 \pm 0.31$ | $7.86 \pm 0.23$ | $6.28 \pm 0.26$ |

## E  USE OF LARGE LANGUAGE MODELS

In accordance with the ICLR 2026 submission policy, we disclose that Large Language Models were used to assist in grammar correction.

