# OpenReview forum: "Lookahead Unmasking Elicits Reliable Decoding in Diffusion Language Models"
_ICLR.cc/2026/Conference — Submitted to ICLR 2026_

### Official Review · Reviewer_nMFR · 2025-10-26

**Soundness:** 2
**Presentation:** 2
**Contribution:** 2
**Rating:** 2
**Confidence:** 5

**Summary:**

The paper proposes **Lookahead Unmasking (LookUM)**, a method to improve inference in masked diffusion models by optimizing the order of token unmasking through a two-step process: a *path generator* that proposes candidate unmasking sets, and a *path verifier* that scores these candidates using uncertainty estimated from one-step-ahead (“lookahead”) predictions. The core idea that I get is that local unmasking errors increase sequence-level uncertainty, and that LookUM can potentially avoid these error-prone trajectories by verifying potential paths before committing to an unmasking order. The paper shows empirical gains across six benchmarks (mathematical reasoning, planning, and code generation) using LLaDA and LLaDA 1.5 models, claiming improved accuracy with minimal computational overhead. The results suggest that uncertainty-based lookahead offers complementary benefits to reinforcement learning–based fine-tuning.

**Strengths:**

- The authors reformulate unmasking as a path selection problem, offering a conceptually clean framing of inference-time decision-making in diffusion language models.

- They demonstrate strong empirical improvements across multiple reasoning tasks (up to +8 points on MBPP, +4 on GSM8K), even over RL-tuned LLaDA 1.5.

- It's computationally efficient — optimal performance achieved with only 2–3 candidate paths, comparable in cost to classifier-free guidance.

- The paper is clearly written and well-organized, with ablation studies separating the impact of each component (path generator, verifier, sampling scheme).

- Demonstrates complementary improvements to existing reinforcement learning–based optimization, suggesting broader applicability.

**Weaknesses:**

1. The claimed theoretical motivation for the verifier is superficial. While the paper suggests that uncertainty correlates with path correctness, there is no formal justification (like via KL optimal control or stochastic path planning theory) linking the proposed verifier to optimal decoding behavior.

2. The definition and construction of the “lookahead state” $\tilde{x}_{t-1}$ isn't well put forth. It is unclear whether this state is sampled deterministically (argmax) or stochastically (categorical sampling), or whether such a proxy meaningfully represents the true next-step state distribution.

3. The verifier’s uncertainty signal is evaluated only using entropy or confidence; there is no evidence is given that this correlates causally with improved unmasking fidelity or that it generalizes beyond the tested benchmarks.

4. GSM8K performance values in Table 2 (≈70s) do not match Figure 3 (≈30), suggesting either metric mismatch or reporting inconsistency?

5. Key hyperparameters (e.g., the threshold in Certainty Filtering, Table 4) are missing, and sensitivity analyses for these thresholds are not reported.
6. The biggest weakness is by far the lack of comparison to **Path Planning (P²)** sampling (Peng et al., 2025), which addresses an almost identical problem—selecting optimal unmasking sequences via planning. As a result, this paper lacks significant novelty.

7. The improvement claims depend entirely on *in silico* reasoning accuracy. There are no wet-lab or real-world experimental validations to show whether uncertainty-guided decoding actually leads to more robust or interpretable outcomes beyond benchmark accuracy.

**Questions:**

1. How is the “lookahead state” $\tilde{x}_{t-1}$ sampled — via argmax, sampling, or another scheme? How sensitive is the verifier’s performance to this choice?

2. Can the authors provide quantitative evidence (with correlation plots or ablation curves) showing that entropy-based uncertainty correlates with true reasoning correctness?

3. What is the threshold used in Certainty Filtering (Table 4), and how does performance vary with different threshold values?

4. How do the authors reconcile the mismatch between GSM8K scores reported in Table 2 and Figure 3?

5. Why haven't the authors compared LookUM to Path Planning (P2) or other planning-based decoding frameworks? If not, can they justify the omission?

6. Could a variant of LookUM integrate model-internal signals (attention entropy, gradient magnitudes, etc.) into the verifier?

7. Beyond benchmark metrics, the authors shoudl perform a wet-lab or grounded validation to test whether uncertainty-based decoding yields outputs that are more interpretable, verifiable, or experimentally meaningful?

With sound responses to these questions, I'd be willing to raise the score to a 4 or even a 6.

---

> ### Comment · Reviewer_nMFR · 2025-11-26
> **Waiting for the authors' response.**
>
> I look forward to receiving the responses! I do think with solid clarifications and comparison to P2, the paper will be in stronger shape and I can increase my score. :)

---

> > ### Author Response · Authors · 2025-11-27
> >
> > Thank you for evaluating LookUM and for providing such helpful and constructive feedback! In the following, we address your feedback and questions in detail, and we look forward to clarifying any remaining points through continued discussion during the rebuttal period.
> >
> > >**W1.** The claimed theoretical motivation for the verifier is superficial. While the paper suggests that uncertainty correlates with path correctness, there is no formal justification (like via KL optimal control or stochastic path planning theory) linking the proposed verifier to optimal decoding behavior.
> >
> >  In the original submission, **we interpreted the sampling process from the perspective of KL-regularized optimal decoding, and Appendix B.3 presented a conceptual link between this objective and the use of SMC/NIS for approximate sampling**. However, we acknowledge that the role of the verifier within this formulation was not described with sufficient clarity. We provide a more explicit explanation below.
> >
> > Reward-aligned sampling denoises from the following steered transition:
> >
> > $p^\ast(x_{t-1}|x_t)\propto p(x_{t-1}|x_t) \exp(r(x_t)/\alpha)$, where  $r(x_t)=\alpha\log \mathbb{E_{x_0 \sim p_\theta(\cdot \mid x_t)}}[\exp (r(x_0)/\alpha)]$.
> >
> > Because $x_0$ is not explicitly available during the discrete diffusion process, we follow standard practice to approximate this quantity using a log–exp first-order approximation:
> >
> > $r(x_t)=\mathbb{E_{x_0 \sim p_\theta(\cdot \mid x_t)}}[r(x_0)]$.
> >
> > Under this formulation, the verifier’s uncertainty-based score becomes a soft value function for the intermediate state. For example:
> >
> > * When using negative entropy as the verifier score:
> >     $\mathbb{E_{x_0 \sim p_\theta(\cdot \mid x_t)}}[r(x_0)] \approx \mathbb{E_{x_0 \sim p_\theta(\cdot \mid x_t)}}[\log p(x_0|x_t)]$,
> >     which corresponds to evaluating the overall uncertainty of the model’s predictive distribution at $x_t$.
> > * When using confidence (maximum probability) as the score:
> >     $\mathbb{E_{x_0 \sim p_\theta(\cdot \mid x_t)}}[r(x_0)] \approx \max_{x_0}  p(x_0|x_t)$,
> >     which assigns higher value to states with more concentrated predictive distributions.
> >
> >     This connection clarifies that LookUM’s candidate selection mechanism implements a  aligns with the theoretical structure of KL-regularized reward-guided decoding. We provide a brief explanation in the main text (L263–L264) and a more detailed discussion in Appendix B.3.
> >
> >
> > >**W2.** The definition and construction of the “lookahead state” $\tilde{x}_{t-1}$ isn't well put forth. It is unclear whether this state is sampled deterministically (argmax) or stochastically (categorical sampling), or whether such a proxy meaningfully represents the true next-step state distribution.
> >
> > >**Q1.** How is  “lookahead state” $\tilde{x}_{t-1}$ sampled — via argmax, sampling, or another scheme? How sensitive is the verifier’s performance to this choice?
> >
> > We thank the reviewer for pointing out this ambiguity. We employ a stochastic procedure for choose the lookahead state, and we describe the details below.
> >
> > **Deterministic vs Stochastic.**
> >
> > The lookahead state $\tilde{x}_{t-1}$ obtained by **unmasking a subset of positions selected from the Certainty Pool $\mathcal{U}_t$**, which is the index set used for candidate expansion. The locations to be unmasked are chosen from $\mathcal{U}_t$, while the tokens placed at these locations are sampled using the **same categorical sampling** procedure employed in the standard MDLM unmasking step. In other words, the lookahead transition follows the model’s predicted token-level distribution, identical to the underlying denoising process, with the only difference being that we apply it selectively to indices within the Certainty Pool.
> >
> >
> > Because the lookahead state is constructed directly from the model’s native predictive distribution, we believe it provides a **meaningful and faithful proxy of the next-step state distribution**, while still enabling controlled exploration through the Certainty Pool.
> >
> > **Sensitivity.**
> >
> >
> > To address the reviewer’s request, we compared the sensitivity differences between the deterministic and stochastic.
> >
> > |      | Humaneval | MATH500 |
> > | -------- | -------- | -------- |
> > | deterministic     | 72.4     | 28.8    |
> > | deterministic     | 72.7      |28.8    |
> >
> > The experimental results show that both approaches yield highly similar outcomes. We believe this behavior arises from the fact that, once the lookahead state $\tilde{x_{t-1}}$
> > is selected as a viable path, both mechanisms ultimately assign $x_{t-1} = \tilde{x_{t-1}}$.
> >
> > We have clarified this construction in the paper L228-L230 for better clarity.

---

> > > ### Author Response · Authors · 2025-11-27
> > >
> > > >**W3.** The verifier’s uncertainty signal is evaluated only using entropy or confidence; there is no evidence is given that this correlates causally with improved unmasking fidelity or that it generalizes beyond the tested benchmarks
> > >
> > > >**Q2.** Can the authors provide quantitative evidence (with correlation plots or ablation curves) showing that entropy-based uncertainty correlates with true reasoning correctness?
> > >
> > > Thank you for the thoughtful question. First, we would like to note that we conducted both qualitative and quantitative analyses to examine whether entropy-based uncertainty correlates with true reasoning correctness. Table 1 in the main paper shows that the model’s entropy increases when an error occurs, and Figure 2 demonstrates, both qualitatively and quantitatively, that leveraging entropy reduces sentence-level reasoning errors. This sentence-level evaluation is necessary because DLMs inherently perform bidirectional unmasking, making it difficult to directly assess reasoning correctness at intermediate states for every denoising step using a process reward model or an LLM-as-a-judge approach.
> > >
> > >
> > > We conducted an additional quantitative analysis on the MATH dataset. For each problem, we generated 10 reasoning paths and accumulated the verifier scores (entropy) selected at each timestep to compute a final aggregated uncertainty score. We then measured the statistical correlation between this score and the correctness of the final answer.
> > >
> > > * Cliff's delta: 0.4930, indicating a large effect size and strong separation between the two distributions.
> > > * Kolmogorov–Smirnov test: KS statistic = 0.3610 with $p < 1.77 \times 10^{-108}$, showing that the distributions differ with extremely high confidence.
> > > * Mann–Whitney U test: $p < 2.70 \times 10^{-130}$, again confirming a strong and statistically significant difference.
> > >
> > > Results demonstrate that the **verifier’s uncertainty score is strongly correlated with true reasoning correctness**, directly addressing the concerns raised in W3 and Q2. The full analysis and figure have been added to Appendix D.2 of the revised manuscript.
> > >
> > >
> > > >**W4.** GSM8K performance values in Table 2 (≈70s) do not match Figure 3 (≈30), suggesting either metric mismatch or reporting inconsistency?
> > >
> > > >**Q4.** How do the authors reconcile the mismatch between GSM8K scores reported in Table 2 and Figure 3?
> > >
> > >  Upon re-examining our figures, we discovered that the captions for the **GSM8K** and **Countdown** curves in Figure 3 were mistakenly swapped, which caused the mismatch the reviewer pointed out. We apologize for this oversight and for any confusion it may have caused.
> > >
> > > We have corrected the caption in the revised version of the paper, and the updated Figure 3 is now fully consistent with the results reported in Table 2. We would like to emphasize that the underlying numerical values were accurate; the issue stemmed solely from mislabeled figure captions.
> > >
> > > We are grateful to the reviewer for bringing this to our attention and helping us improve the clarity and accuracy of our presentation.
> > >
> > >
> > > >**W5.** Key hyperparameters (e.g., the threshold in Certainty Filtering, Table 4) are missing, and sensitivity analyses for these thresholds are not reported.
> > >
> > > >**Q3.** What is the threshold used in Certainty Filtering (Table 4), and how does performance vary with different threshold values?
> > >
> > > During our study, we performed a sweep over Certainty Filtering thresholds, ranging from 0.9 to 0.1 in increments of 0.1, to determine a suitable operating point. In Table 4, we report the results using a threshold of 0.6. In response to the reviewer's request, we provide the full threshold sensitivity results below.
> > >
> > > | |0.9 | 0.8 |0.7|0.6|0.5|0.4|0.3|0.2|0.1|
> > > | -------- | -------- | -------- | -------- |--|--|--|--|--|--|
> > > | GSM8K     | 68.3    | 69.1     | 70.1 |72.6|70.1|70.1|69.2|67.5|67.5|
> > > | MATH500    | 27.4    | 27.6     | 26.4 |26.4|26.1|26.2|26.2|24.4|23.1|
> > > | Countdown     |   20.7  | 20.7     | 20.5 |19.5|21.1|23.1|23.1|20.3|20.1|
> > >
> > > The results show that the optimal threshold varies across tasks, while consistently demonstrating that very low thresholds lead to degraded performance. These findings have been incorporated into the revised manuscript as part of Appendix D.1.

---

> > > > ### Author Response · Authors · 2025-11-27
> > > >
> > > > >**W6.** The biggest weakness is by far the lack of comparison to Path Planning (P2) sampling (Peng et al., 2025), which addresses an almost identical problem—selecting optimal unmasking sequences via planning. As a result, this paper lacks significant novelty.
> > > >
> > > > >**Q5.** Why haven't the authors compared LookUM to Path Planning (P2) or other planning-based decoding frameworks? If not, can they justify the omission?
> > > >
> > > > Thank you very much for raising this important point. We would like to clarify that our core contribution is not in newly asserting the importance of the unmasking path selection itself. Rather, our aim is to formalize the process of selecting unmasking positions within a general reward-guided generation framework, using unmasking path  and verifier scores.
> > > >
> > > > More concretely, P2 integrates various unmasking strategies and remasking behaviors into a path-planner–based denoising process. In P2, the current state $x_t$ is used to obtain the denoiser’s prediction of $x_0$, after which the planner determines which positions to remask and unmask. In this setup, the unmasking method is same to existing greedy strategies. In our understanding, P2’s key contribution lies in providing a unified theoretical framework for combining remasking and unmasking decisions, and in enabling more effective selection of remasking positions.
> > > >
> > > > In contrast, LookUM does not attempt to design a new planning mechanism. Instead, it constructs the proposal pool using existing path-planning method such as confidence, margin, or entropy, and then uses the verifier’s uncertainty based score to guide path selection. Thus, we view LookUM’s primary contribution as recasting the unmasking position selection problem through the lens of reward guided generation with verifier score and unmasking path.
> > > >
> > > > Following the reviewer’s suggestion, we additionally conducted a quantitative comparison against P2 during the rebuttal period. While the original P2 paper did not include LLaDA experiments, we found the authors’ official LLaDA codebase and used it as-is. We performed a search over the key hyperparameter $\eta$ from 0 to 2 with a step size of 0.2, but unfortunately were unable to observe performance improvements under any configuration. The corresponding results are summarized in the table below.
> > > >
> > > > | Data\ $\eta$|Confidence Baseline|0.0 | 0.2 | 0.4 | 0.6 | 0.8 | 1.0 | 1.2 | 1.4 | 1.6 | 1.8 | 2.0|
> > > > | -------- | -------- | -------- | -------- | -------- | -------- |-------- | -------- | -------- | -------- | -------- |-------- |-------- |
> > > > | GSM8K   |   **72.7** |   58.7   |   66.6   |66.0 |   65.8   |65.3 |   66.0   |  66.2  |    66.9   |67.2 |   67.4   | 67.4|
> > > > | MATH   |   **28.8**  |   21.0   |   23.6   | 25.0|   22.4   |22.6 |   22.4   |  23.8  |    23.6   | 24.6|  24.2    |24.2|
> > > >
> > > > Nonetheless, we believe that P2 makes an important contribution by establishing a theoretical foundation for remasking/unmasking selection via an expanded ELBO, and we have incorporated a more detailed discussion of this perspective in the revised Related Work section. We have also clarified the language in lines L91–L92,L197-L199 to better convey that LookUM’s main contribution lies not in the path-selection problem itself, but in reframing unmasking-position selection within a verifier-based, reward-guided generation framework.

---

> > > > > ### Author Response · Authors · 2025-11-27
> > > > >
> > > > > > **W7.** The improvement claims depend entirely on in silico reasoning accuracy. There are no wet-lab or real-world experimental validations to show whether uncertainty-guided decoding actually leads to more robust or interpretable outcomes beyond benchmark accuracy.
> > > > >
> > > > > >**Q7.** Beyond benchmark metrics, the authors shoudl perform a wet-lab or grounded validation to test whether uncertainty-based decoding yields outputs that are more interpretable, verifiable, or experimentally meaningful?
> > > > >
> > > > > First, we would like to respectfully clarify that our work falls within the domain of language diffusion models, where benchmark-based evaluations are widely regarded as the standard and most reliable approach for assessing reasoning quality.
> > > > >
> > > > > In our experiments, we consistently observed that LookUM improves performance across diverse task categories, including coding tasks where **full-sequence accuracy is essential**, mathematical **reasoning tasks that require precise answers**, and planning tasks where the **denoising order plays a critical role**. We believe that LookUM’s consistent improvements on structurally diverse datasets demonstrate its generalization capability.
> > > > >
> > > > > In addition, **Section 3.1 includes an LLM-based verifier analysis that provides a more interpretable perspective on how LookUM corrects local reasoning errors during decoding,** beyond merely reporting final accuracy.
> > > > >
> > > > > That said, it is possible that we may not be fully understanding the reviewer’s intended meaning regarding **“wet-lab or grounded validation”.** Since Language Diffusion Models do not interact with physical environments, we would be grateful if the reviewer could kindly clarify what specific form of grounded or experimental validation they have in mind. If there is a particular type of evaluation you believe would meaningfully strengthen our work, we would be more than happy to consider incorporating it.
> > > > >
> > > > >
> > > > > >**Q6.** Could a variant of LookUM integrate model-internal signals (attention entropy, gradient magnitudes, etc.) into the verifier?
> > > > >
> > > > > Thank you for raising this thoughtful point regarding the extensibility of the verifier.
> > > > > In response to your question, we conducted two additional variants of LookUM that integrate internal model signals into the verifier:
> > > > >
> > > > > * **Attention Entropy**: We used the entropy of the final attention layer as the uncertainty score.
> > > > > * **Gradient Magnitudes**:
> > > > > We computed the gradient norm of the model’s prediction logits and used it as the score.
> > > > >
> > > > > These experiments were carried out under the same evaluation setup as our main results, and the outcomes are summarized in the table below:
> > > > >
> > > > >
> > > > > |          | MATH500 | GSM8K |Countdown |
> > > > > | -------- | -------- | -------- |-------- |
> > > > > | LookUM     | 28.8     | 72.7     |25.4     |
> > > > > | Cofindence     | 26.0     | 68.3     |20.3     |
> > > > > | Attention Entropy     | 27.2     | 70.5     |23.1     |
> > > > > | Gradient Magnitudes     | 26.4     | 68.4     |23.5     |
> > > > >
> > > > > Both variants showed lower performance than the original LookUM, but still outperformed confidence-based decoding. These results suggest that both attention entropy and gradient magnitudes have the potential to identify promising decoding paths. We believe that integrating internal model signals remains a promising research direction, and your suggestion provides meaningful insight into the extensibility of verifier-based approaches.
> > > > >
> > > > > We sincerely appreciate the reviewer’s insightful questions and constructive suggestions.
> > > > > They helped us clarify key aspects of our method and significantly improved the completeness and rigor of the revised manuscript.
> > > > > Thank you again for your thoughtful and detailed feedback.

---

> > > > > > ### Comment · Reviewer_nMFR · 2025-11-27
> > > > > >
> > > > > > Thank you for the detailed and thoughtful responses, and for clarifying my misunderstandings! After going through the rebuttal and the revised explanations, I am satisfied that most of my original concerns have been addressed.
> > > > > >
> > > > > > The clarifications you provided were helpful. The added explanation of how the verifier fits into the KL-guided decoding view makes the role of uncertainty much clearer, and the construction of the lookahead state is now understandable to me. Also, the quantitative check on entropy and correctness was good to see. The fixes to the GSM8K figure, the threshold discussion, and the comparison to P2 resolve the main points I was worried about. I also appreciate the extra experiments the authors ran, even the ones that did not help!
> > > > > >
> > > > > > Btw, I did not explicitly see the revisions incorporated into the updated manuscript. Assuming the revised manuscript integrates these changes as described, I am raising my score to a 6.

---

> > > > > > > ### Author Response · Authors · 2025-11-27
> > > > > > >
> > > > > > > Thank you again for updating the score and for your continued engagement !  We have uploaded the revised version accordingly.
> > > > > > >
> > > > > > > If there is anything that would benefit from further clarification during the remaining rebuttal period, please let us know. Your feedback has been invaluable in improving the paper.

---

### Official Review · Reviewer_Labt · 2025-10-30

**Soundness:** 2
**Presentation:** 3
**Contribution:** 2
**Rating:** 4
**Confidence:** 4

**Summary:**

This paper introduces Lookahead Unmasking (LookUM), an inference-time decoding framework for masked diffusion language models (MDLMs). The method reframes unmasking as a path selection problem: at each step, multiple unmasking “paths” are proposed, and a verifier based on sequence-level uncertainty selects the most consistent one. The approach is model-agnostic, requires no fine-tuning, and achieves consistent performance gains on reasoning, coding, and planning benchmarks with LLaDA and LLaDA-1.5.

**Strengths:**

The paper is well-written, clear, and motivated by a relevant problem — improving diffusion model decoding efficiency.

The formulation of unmasking as path selection is intuitive and offers a unifying framework that could, in principle, incorporate uncertainty, reward, or heuristic guidance.

The experimental setup is extensive, covering multiple reasoning and coding benchmarks and both base and RL-tuned diffusion LMs.

**Weaknesses:**

#### **1. Unfair comparison due to unequal compute budgets**

The main experimental results (e.g., Table 2 ) compare LookUM — which explicitly samples **multiple paths (2–4 per step)** — against baseline methods that only use **a single sampling trajectory**. This means that LookUM’s inference-time compute is **2–4× higher**, as each path requires a separate forward pass through the model .
The authors claim that “the verifier overhead is negligible,” but the dominant cost in MDLM inference is model evaluation itself, not verifier scoring. Thus, a method using (k) paths incurs roughly (k\times) inference cost. Comparing multi-path results against single-path baselines is **not a fair comparison** of sampling quality per unit compute.

For example, if LookUM achieves higher accuracy on HumanEval or GSM8K using 2 paths, it is effectively performing twice the work. The proper control would be either (a) match compute (e.g., let baselines sample twice), or (b) report performance *per unit of FLOPs* or wall-time. Without this normalization, the empirical improvement is difficult to interpret.

#### **2. Comparison with Baselines**

The conceptual framework of treating unmasking as *path planning or path selection* has already appeared in earlier diffusion-decoding work. Prior studies (e.g., those exploring **path-planning for masked diffusion models** and **
Train for the Worst, Plan for the Best: Understanding Token Ordering in Masked Diffusions**.  The authors should compare with them.

**Questions:**

See Weaknesses.

---

> ### Author Response · Authors · 2025-11-27
>
> Thank you for evaluating LookUM and for providing such helpful and constructive feedback. We are especially grateful for your insightful suggestion regarding the comparison methodology, which has been extremely valuable in refining our experimental evaluation. In the following, we address your feedback and questions in detail, and we look forward to clarifying any remaining points through continued discussion during the rebuttal period.
>
> > W1. Unfair comparison due to unequal compute budgets
>
> Thank you very much for raising this important point regarding compute-matched comparisons. Before presenting the results, we would like to clarify that our method assumes sufficient inference-time compute, and thus strict compute matching with baselines is not always required. Nevertheless, following the reviewer’s suggestions, we performed both (a) matched-compute evaluations and (b) wall-clock time measurements.
>
>
>
>
> First, following the reviewer’s suggestion(a), we conducted compute-matched evaluations. Since LookUM uses at most three paths in the main experiments, we report the baselines’ performance under the same 3-path setting. For convenience, we summarize the key results below.
>
> These additional results show that simply **increasing the number of paths is not an effective strategy, and that LookUM remains competitive even when compared under the same inference budget**.
>
> | Model        | Method                  | GSM8K-128 | GSM8K-256 | MATH500-128 | MATH500-256 | Countdown-128 | Countdown-256 |
> |--------------|--------------------------|-----------|-----------|--------------|--------------|-----------|-----------|
> | **LLaDA**    | Confidence (×1)          | 68.3      | 76.7      | 26.0         | 32.4         | 20.3      | 21.9      |
> |              | Confidence (×3)          | 68.8      | 77.7      | 22.8         | 28.8         | 19.5      | 12.1      |
> |              | Margin (×1)              | 67.1      | 76.1      | 28.4         | 34.4         | 19.1      | 20.7      |
> |              | Margin (×3)              | 70.6      | 77.1      | 23.4         | 30.2         | 17.3      | 11.3      |
> |              | Entropy (×1)             | 66.7      | 75.4      | 26.0         | 33.0         | 21.9      | 20.3      |
> |              | Entropy (×3)             | 59.6      | 69.9      | 22.2         | 27.8         | 13.4      | 10.2      |
> |              | PC-Sampler (×1)          | 67.3      | 73.7      | 25.2         | 32.4         | **26.5**  | 20.3      |
> |              | PC-Sampler (×3)          | 66.0      | 75.0      | 21.8         | 27.6         | 14.5      | 12.0      |
> |              | ReMDM                    | 69.1      | 77.9      | 27.4         | 33.0         | 25.3      | 17.2      |
> |              | **LookUM**               | **72.7**  | **79.3**  | **28.8**     | **34.6**     | 25.4      | **23.1**  |
> | **LLaDA-1.5**| Confidence (×1)          | 69.5      | 79.4      | 28.6         | 32.6         | 20.3      | 23.4  |
> |              | Confidence (×3)          | 71.1      | 79.7      | 27.3         | 29.2         | 20.1      | **23.5**     |
> |              | Margin (×1)              | 71.3      | 78.3      | 27.2         | 35.0     | 24.6      | 14.0      |
> |              | Margin (×3)              | 71.4      | 78.4      | 24.4         | 30.6         | 24.6      | 14.1      |
> |              | Entropy (×1)             | 69.7      | 77.0      | 28.2         | 32.2         | 23.0      | 12.9      |
> |              | Entropy (×3)             | 71.1      | 80.1      | 22.6         | 29.8         | 15.1      | 12.7      |
> |              | PC-Sampler (×1)          | 70.1      | 77.3      | 26.6         | 32.2         | 25.4      | 19.1      |
> |              | PC-Sampler (×3)          | 69.9      | 79.5      | 22.6         | 34.4         | 19.8      | 14.5      |
> |              | ReMDM                    | 70.4      | 80.1      | 27.4         | 34.0         | 23.4      | 19.9      |
> |              | **LookUM**               | **74.5**  | **82.3**  | **29.2**     | **35.8**     | **27.3**  | 17.9      |

---

> ### Author Response · Authors · 2025-11-27
>
> In addition, following point (b), we also conducted wall-clock time measurements. The table below summarizes the execution time of each method under identical conditions.
>
> | Setting     |LookUM  |ReMDM          |       Entropy | Confidence | Margin  | PC-sampler    |
> | ----------- | ------------- | ---------------- | ------------- | ------------- | -------------- | ------------- |
> | **1 path**  |–              | –             | 4.343 ± 0.208 | 4.303 ± 0.171    | 4.527 ± 0.369 | 4.270 ± 0.154 |
> | **3 paths** | 7.337 ± 0.424 | 16.697 ± 0.212 | 7.074 ± 0.485 | 6.417 ± 0.307    | 7.860 ± 0.225 | 6.283 ± 0.257 |
>
> The results show that LookUM efficiently maintains low wall-clock time by processing k paths in parallel, even with higher theoretical compute. In contrast, ReMDM performs sequential remasking and therefore exhibits much longer execution time. This demonstrates that LookUM benefits greatly from parallelization and achieves superior practical efficiency.
>
> At the same time, we would like to clarify the intended
> scope of our work. Our primary research question focuses on how to most effectively utilize inference-time computational budget given a fixed maximum number of denoising steps. Importantly, diffusion language models differ from conventional autoregressive LLMs in that many common scaling strategies cannot be applied. For example:
> * **Classifier-Free Guidance (CFG)** is essentially infeasible due to the nature of the discrete language probability space.
> * As shown in Section 4.3, **reward-model scoring does not function** reliably because intermediate diffusion states remain highly noisy, making it difficult for external evaluators to produce meaningful gradients or scores.
> * Most critically, increasing the number of denoising steps is not an available axis for improvement, since the step count is inherently capped by the sequence length and the formulation of diffusion-based language tokenization.
>
> In other words, unlike standard LLMs or continuous image diffusion models, **diffusion language models cannot scale performance by simply expanding inference-time computation**. This structural limitation motivates our focus: LookUM aims not to minimize cost per sample, but rather to provide a principled and efficient strategy for leveraging additional computation within a fixed step budget by exploring more informative paths during test-time inference.
>
> The additional experimental results have been incorporated into Table 2 of the revised manuscript. Further explanations regarding the baselines have been added in lines L325–L328, and the discussion on LookUM’s efficient cost scaling has been included in L432–L439.
>
> >W2. Comparison with Baselines
>
> We sincerely thank the reviewer for requesting a comparison with prior work. We view this as an important opportunity to further clarify the contributions of our paper. To clarify, **LookUM does not claim novelty in treating unmasking as path planning or path selection**. Our contribution is in **formalizing unmasking position selection within a reward guided generation framework using verifier scores and sampled paths**.
>
> We conducted both qualitative and quantitative analyses comparing our method with two relevant studies and incorporated these comparisons into the Related Works. For convenience, we summarize the key points below.
>
>
> 1. Quantitative Comparison
> * Train for the Worst
> **The decoding method(margin) in paper was already included in our main experiments(Figure 2)**, and we observed that LookUM consistently outperforms the margin-based approach.
> * Path-Planning for Masked Diffusion Models Sampling
> Although the original P2 paper does not include LLaDA experiments, the authors provide an official LLaDA implementation, which we used to apply the P2 sampler. However, even when following the same configuration, the sampler did not produce usable results.
>
> We conducted a broad hyperparameter search over the key parameter $\eta$, following the paper’s specification of sweeping from 0 to 2 in increments of 0.2. However, **across all settings, the P2 sampler consistently showed decreased performance rather than improvements**. The full search results are summarized in the table below.
>
>
> | Data\ $\eta$|LookUM|0.0 | 0.2 | 0.4 | 0.6 | 0.8 | 1.0 | 1.2 | 1.4 | 1.6 | 1.8 | 2.0|
> | -------- | -------- | -------- | -------- | -------- | -------- |-------- | -------- | -------- | -------- | -------- |-------- |-------- |
> | GSM8K   |   **72.7** |   58.7   |   66.6   |66.0 |   65.8   |65.3 |   66.0   |  66.2  |    66.9   |67.2 |   67.4   | 67.4|
> | MATH   |   **28.8**  |   21.0   |   23.6   | 25.0|   22.4   |22.6 |   22.4   |  23.8  |    23.6   | 24.6|  24.2    |24.2|

---

> > ### Author Response · Authors · 2025-11-27
> >
> > 2. Qualitative Comparison
> > * Train for the Worst
> > This work discusses the computational challenges of training masked diffusion models and proposes a margin-based decoding strategy at inference time. However, it does not explore decoding strategies that consider multiple generative paths. Its approach selects a single decoding path based on prediction margins, which differs fundamentally from the LookUM.
> >
> > * Path-Planning for Masked Diffusion Models Sampling
> > P2 examines how unmasking order affects generation quality and introduces a path-planning perspective. However, P2’s primary contribution lies in its remasking mechanism; it does not propose a mechanism for improving the quality of unmasking. P2 retains the standard decoding rule during unmasking, whereas LookUM introduces a guided generation with verifier score. We respectfully emphasize that LookUM is therefore orthogonal and complementary to P2.
> >
> > We have incorporated this discussion into the revised Related Work section, and we have updated lines L91-L92 and L197-L199 to emphasize that our contribution lies in a reward guided generation framework that uses verifier scores and sampled paths.

---

### Official Review · Reviewer_6sEz · 2025-11-01

**Soundness:** 2
**Presentation:** 3
**Contribution:** 2
**Rating:** 4
**Confidence:** 3

**Summary:**

This paper introduces Lookahead Unmasking (LookUM) for diffusion language models, reframing decoding as path selection over unmasking orders. A path generator samples candidate unmasking sets from a high-certainty pool, and a verifier scores one-step lookahead states using sequence-level uncertainty (avg. negative entropy or confidence), selecting paths via importance sampling (SMC/NIS). LookUM yields consistent gains on several reasoning benchmarks.

**Strengths:**

- The proposed method is conceptually neat and easy to plug into existing pretrained diffusion language models. It's also a good combination of search algorithms and diffusion language models.
- The proposed method consistently improves performance on several reasoning benchmarks (math, code, sudoku) compared with recent baselines.
- This paper conducts detailed ablation studies on components of the proposed method and explores the integration with external reward models.

**Weaknesses:**

- While the proposed method shows improvements on benchmarks, the score difference compared to the best baselines is not very large, especially given that the proposed method's computational cost is 2-3$\times$ as much.
- This paper doesn't show or compare measurements on the actual inference cost. That would make the performance-cost trade-off clearer.
- The number of lookahead steps is an important hyperparameter for the proposed method. Throughout the paper it's fixed to be $1$. Why not consider more lookahead steps (e.g., more candidate branches, fewer lookahead steps VS fewer candidate branches, more lookahead steps, under the same computational budget)? What could be the difficulties?

**Questions:**

Please see Weaknesses

---

> ### Author Response · Authors · 2025-11-27
>
> We deeply appreciate the reviewer for constructive comments and helpful feedback. In the following, we carefully address your questions, and hope this fully resolves your concerns.
>
>
> >W1. While the proposed method shows improvements on benchmarks, the score difference compared to the best baselines is not very large, especially given that the proposed method's computational cost is 2-3 as much.
>
> Thank you very much for this constructive comment. We agree that the absolute score differences may appear modest at first glance. We believe that the consistent and stable performance **improvements observed in our work represent a genuinely meaningful advancement**. In addition, we would like to emphasize that, in **language diffusion models, simply increasing inference time does not translate into better performance**.
>
>
> For example, the 8% improvement on MBPP corresponds to roughly a **28% relative gain** over the best existing baseline, a practically meaningful jump that brings our method into the performance range typically achieved by RL-tuned models, even though it requires no additional training.
>
>  In addition, despite using the same inference cost, ReMDM shows a considerable performance gap compared to LookUM, and our method also yields results that remain competitive with LLaDA-1.5, which is further post-trained through reinforcement learning.
>
> Therefore, even if the absolute score differences may appear modest, the fact that LookUM consistently improves performance across diverse domains and evaluation settings provides strong evidence of its robust generalization capabilities.
>
> Additionally, **we conducted multi-sampling experiments in which all baseline methods were assigned the same inference budget as LookUM**, using three candidate paths since LookUM employs at most three paths in the main experiments. Accordingly, we conducted our evaluation of both LLaDA and LLaDA-1.5 under the same setting, using exactly three paths for all results. The experiments were conducted on GSM8K, MATH, and Countdown, which each provide a single comparable ground-truth answer. Evaluation was performed by treating equivalent expressions as identical.
>
> | Model        | Method                  | GSM8K-128 | GSM8K-256 | MATH500-128 | MATH500-256 | Countdown-128 | Countdown-256 |
> |--------------|--------------------------|-----------|-----------|--------------|--------------|-----------|-----------|
> | **LLaDA**    | Confidence (×1)          | 68.3      | 76.7      | 26.0         | 32.4         | 20.3      | 21.9      |
> |              | Confidence (×3)          | 68.8      | 77.7      | 22.8         | 28.8         | 19.5      | 12.1      |
> |              | Margin (×1)              | 67.1      | 76.1      | 28.4         | 34.4         | 19.1      | 20.7      |
> |              | Margin (×3)              | 70.6      | 77.1      | 23.4         | 30.2         | 17.3      | 11.3      |
> |              | Entropy (×1)             | 66.7      | 75.4      | 26.0         | 33.0         | 21.9      | 20.3      |
> |              | Entropy (×3)             | 59.6      | 69.9      | 22.2         | 27.8         | 13.4      | 10.2      |
> |              | PC-Sampler (×1)          | 67.3      | 73.7      | 25.2         | 32.4         | **26.5**  | 20.3      |
> |              | PC-Sampler (×3)          | 66.0      | 75.0      | 21.8         | 27.6         | 14.5      | 12.0      |
> |              | ReMDM                    | 69.1      | 77.9      | 27.4         | 33.0         | 25.3      | 17.2      |
> |              | **LookUM**               | **72.7**  | **79.3**  | **28.8**     | **34.6**     | 25.4      | **23.1**  |
> | **LLaDA-1.5**| Confidence (×1)          | 69.5      | 79.4      | 28.6         | 32.6         | 20.3      | 23.4  |
> |              | Confidence (×3)          | 71.1      | 79.7      | 27.3         | 29.2         | 20.1      | **23.5**     |
> |              | Margin (×1)              | 71.3      | 78.3      | 27.2         | 35.0     | 24.6      | 14.0      |
> |              | Margin (×3)              | 71.4      | 78.4      | 24.4         | 30.6         | 24.6      | 14.1      |
> |              | Entropy (×1)             | 69.7      | 77.0      | 28.2         | 32.2         | 23.0      | 12.9      |
> |              | Entropy (×3)             | 71.1      | 80.1      | 22.6         | 29.8         | 15.1      | 12.7      |
> |              | PC-Sampler (×1)          | 70.1      | 77.3      | 26.6         | 32.2         | 25.4      | 19.1      |
> |              | PC-Sampler (×3)          | 69.9      | 79.5      | 22.6         | 34.4         | 19.8      | 14.5      |
> |              | ReMDM                    | 70.4      | 80.1      | 27.4         | 34.0         | 23.4      | 19.9      |
> |              | **LookUM**               | **74.5**  | **82.3**  | **29.2**     | **35.8**     | **27.3**  | 17.9      |

---

> > ### Author Response · Authors · 2025-11-27
> >
> > The results show that even **under equal computational cost, LookUM consistently outperforms all baselines**. This further demonstrates that existing unmasking strategies lack a structural mechanism to convert additional test-time compute into meaningful performance gains. In other words, **simply increasing the number of samples does not improve their performance**, revealing a fundamental limitation of these approaches.
> >
> > The updated experimental results have been added to Table 2 of the revised manuscript, and the discussion regarding LookUM’s performance has been included in  L432–L439.
> >
> > >W2. This paper doesn't show or compare measurements on the actual inference cost. That would make the performance-cost trade-off clearer.
> >
> > Thank you for the insightful feedback. Following your suggestion, we conducted an additional wall-clock time analysis to more clearly compare the actual inference cost.
> >
> > Since LookUM uses up to three paths in our main experiments, we evaluated all baseline methods under both single-path and 3-path settings to ensure a fair comparison. The table below summarizes the execution time measured under identical conditions.
> >
> > | Setting     |LookUM  |ReMDM          |       Entropy | Confidence | Margin  | PC-sampler    |
> > | ----------- | ------------- | ---------------- | ------------- | ------------- | -------------- | ------------- |
> > | **1 path**  |–              | –             | 4.343 ± 0.208 | 4.303 ± 0.171    | 4.527 ± 0.369 | 4.270 ± 0.154 |
> > | **3 paths** | 7.337 ± 0.424 | 16.697 ± 0.212 | 7.074 ± 0.485 | 6.417 ± 0.307    | 7.860 ± 0.225 | 6.283 ± 0.257 |
> >
> > The results show that LookUM can process k paths in parallel on the GPU, allowing it to maintain efficient wall-clock time despite the increased theoretical computation. In contrast, ReMDM performs remasking sequentially at every step, which leads to significantly longer execution time even when the theoretical cost is similar. These findings indicate that, despite its higher nominal computational cost, **LookUM benefits substantially from parallelization** and achieves superior practical efficiency during inference.
> >
> > The explanation of inference cost added to Appendix D.3 in the revised manuscript.
> >
> >
> >
> > >W3. The number of lookahead steps is an important hyperparameter for the proposed method. Throughout the paper it's fixed to be 1. Why not consider more lookahead steps (e.g., more candidate branches, fewer lookahead steps VS fewer candidate branches, more lookahead steps, under the same computational budget)? What could be the difficulties?
> >
> >
> > We appreciate the reviewer for raising this important point regarding the choice of lookahead depth. During development, we also experimented with larger lookahead depths, and consistently observed that increasing the depth excessively leads to performance degradation.
> >
> > However, we view this not as a limitation, but rather as evidence that the verifier already provides a highly accurate 1-step estimate. The verifier score approximates the expected reward of terminal states, and while deeper lookahead can compute a more “precise” prediction along a single rollout, such estimates exhibit high variance and fail to faithfully represent next states $x_{t-1}$.
> >
> > In other words, this observation indicates that a 1-step lookahead already supplies sufficiently accurate guidance, making longer roll-outs not cost-effective.
> >
> > We report below the results obtained by increasing the number of lookahead steps while keeping the underlying path fixed.
> > |  | GSM8K | MATH500 |
> > | -------- | -------- | -------- |
> > | LookUM     | 72.7     |   28.8   |
> > |    2 step | 72.2     |   27.6   |
> > | 3 step     | 70.0    |   25.8   |
> > | 4 step     | 69.5     |   25.8   |
> >
> >
> > Thank you for raising important questions. We hope that our response satisfactorily addresses the reviewer’s concerns.

---

### Official Review · Reviewer_FvKa · 2025-11-04

**Soundness:** 3
**Presentation:** 3
**Contribution:** 2
**Rating:** 6
**Confidence:** 4

**Summary:**

Masked Diffusion Models (MDMs) train with an any-order objective, allowing multiple possible sampling paths. This paper addresses the problem of finding an optimal unmasking path during inference. Existing heuristic strategies are typically locally greedy and fail to capture sequence-level dependencies. To address this, the authors propose Lookahead Unmasking (LookUM) — a method that uses the average uncertainty of the next step to guide path selection. LookUM reframes decoding as a path selection problem, consisting of two components: a path generator that proposes candidate unmasking paths, and a verifier that scores these paths using sequence-level uncertainty. The paper reports strong empirical results.

**Strengths:**

- The paper proposes a simple yet effective inference-time approach for identifying optimal unmasking paths without modifying the training process.
- The method is evaluated across multiple benchmarks and achieves strong empirical performance compared to existing baselines.

**Weaknesses:**

- While LookUM increases inference-time computation, the paper does not clearly quantify this overhead compared to baseline methods. A more detailed analysis of inference time should be included in the paper.
- Another way to improve diffusion model performance under a higher inference-time budget is to increase the number of denoising steps. It would be interesting to compare LookUM against this baseline under a fixed compute or time budget, to better understand its effectiveness.
- The approach performs multiple forward passes for each inference step. Although conceptually simple, this can substantially increase inference time. However, it might not be necessary to perform LookUM during each of the inference steps. It would be useful to explore whether using any method could decide when to apply lookahead unmasking — potentially avoiding unnecessary lookahead steps and reducing the inference cost without significant performance loss.

**Questions:**

- Would LookUM perform better if the verifier did not average negative entropy over all tokens? Perhaps focusing on a subset of the most uncertain tokens could reduce noise in the uncertainty estimation.
- Is there any intuitive reason of the performance drop as we increase the number of paths (e.g., in MATH500)? How frequently does it occur? One possible explanation is that the estimated uncertainty does not perfectly correlate with the true path quality.

---

> ### Author Response · Authors · 2025-11-27
>
> We sincerely appreciate the reviewer's constructive comments and positive feedback on our manuscript.
>
> > W1. While LookUM increases inference-time computation, the paper does not clearly quantify this overhead compared to baseline methods. A more detailed analysis of inference time should be included in the paper.
>
> Thank you for the constructive feedback. Although Section 3.3 of the original manuscript discusses the computational complexity, we agree that a more concrete explanation is necessary. Below, we explain detail of the computational cost of LookUM and its difference in actual wall-clock time.
>
> Baseline methods (confidence, margin, entropy, PC-sampler) perform a total of T model calls over T timesteps. For fair comparison, ReMDM was configured to match the number of model calls used by LookUM.
>
> The computational cost of LookUM can be divided into two parts:
> * Model calls: LookUM performs exactly $T\times k$ model calls, where
> $T$ is the number of timesteps and $k$ is the number of paths. The verifier score reuses the same prediction, so no additional model calls are required.
>
> * Additional operations: The pooling step, which computes confidence for candidate tokens and selects the top-k , is also performed in the baselines and does not introduce any additional overhead specific to LookUM. Therefore, the computational cost associated with this step is likewise $k$ times that of the baselines.
>
>
> Consequently, the total computational cost becomes almost exactly k times that of the baseline.
>
> We additionally measured the wall-clock time to provide a practical comparison. Since LookUM uses at most 3 paths in the main experiments, we report measurements based on 3-path decoding. For a clearer comparison, we also measured the time required for the baseline methods to generate 3 paths under the same setting. The results are summarized below:
>
>
> | Setting     |LookUM  |ReMDM          |       Entropy | Confidence | Margin  | PC-sampler    |
> | ----------- | ------------- | ---------------- | ------------- | ------------- | -------------- | ------------- |
> | **1 path**  |–              | –             | 4.343 ± 0.208 | 4.303 ± 0.171    | 4.527 ± 0.369 | 4.270 ± 0.154 |
> | **3 paths** | 7.337 ± 0.424 | 16.697 ± 0.212 | 7.074 ± 0.485 | 6.417 ± 0.307    | 7.860 ± 0.225 | 6.283 ± 0.257 |
>
>
> The results show that **LookUM can process k paths in parallel on the GPU, allowing it to maintain efficient wall-clock** time despite the increased computational cost. In contrast, ReMDM performs sequential remasking at every step, which leads to substantially longer execution time in practice.
>
> In the revised manuscript, we have added an explicit explanation of the computational overhead in  L304–L305, and we now report the overhead of each method in Table 2.

---

> > ### Author Response · Authors · 2025-11-27
> >
> > >W2. Another way to improve diffusion model performance under a higher inference-time budget is to increase the number of denoising steps. It would be interesting to compare LookUM against this baseline under a fixed compute or time budget, to better understand its effectiveness.
> >
> >
> > Unlike continuous diffusion models, diffusion language models have inherent limitations in scaling through timesteps: The maximum number of steps is bounded by sequence length, as each step must unmask at least one token.
> >
> > Given these fundamental limitations, we conduct experiments where baseline methods generate multiple candidates under equivalent computational budgets, using three candidate paths since LookUM employs at most three paths in the main experiments. In the main experiments, LookUM using at most three candidate paths. Accordingly, we conducted our evaluation of both LLaDA and LLaDA-1.5 under the same setting, using exactly three paths for all results. The experiments were conducted on GSM8K, MATH, and Countdown, which each provide a single comparable ground-truth answer. Evaluation was performed by treating equivalent expressions as identical.
> >
> >
> > | Model        | Method                  | GSM8K-128 | GSM8K-256 | MATH500-128 | MATH500-256 | Countdown-128 | Countdown-256 |
> > |--------------|--------------------------|-----------|-----------|--------------|--------------|-----------|-----------|
> > | **LLaDA**    | Confidence (×1)          | 68.3      | 76.7      | 26.0         | 32.4         | 20.3      | 21.9      |
> > |              | Confidence (×3)          | 68.8      | 77.7      | 22.8         | 28.8         | 19.5      | 12.1      |
> > |              | Margin (×1)              | 67.1      | 76.1      | 28.4         | 34.4         | 19.1      | 20.7      |
> > |              | Margin (×3)              | 70.6      | 77.1      | 23.4         | 30.2         | 17.3      | 11.3      |
> > |              | Entropy (×1)             | 66.7      | 75.4      | 26.0         | 33.0         | 21.9      | 20.3      |
> > |              | Entropy (×3)             | 59.6      | 69.9      | 22.2         | 27.8         | 13.4      | 10.2      |
> > |              | PC-Sampler (×1)          | 67.3      | 73.7      | 25.2         | 32.4         | **26.5**  | 20.3      |
> > |              | PC-Sampler (×3)          | 66.0      | 75.0      | 21.8         | 27.6         | 14.5      | 12.0      |
> > |              | ReMDM                    | 69.1      | 77.9      | 27.4         | 33.0         | 25.3      | 17.2      |
> > |              | **LookUM**               | **72.7**  | **79.3**  | **28.8**     | **34.6**     | 25.4      | **23.1**  |
> > | **LLaDA-1.5**| Confidence (×1)          | 69.5      | 79.4      | 28.6         | 32.6         | 20.3      | **23.4**  |
> > |              | Confidence (×3)          | 71.1      | 79.7      | 27.3         | 29.2         | 20.1      | 23.5      |
> > |              | Margin (×1)              | 71.3      | 78.3      | 27.2         | **35.0**     | 24.6      | 14.0      |
> > |              | Margin (×3)              | 71.4      | 78.4      | 24.4         | 30.6         | 24.6      | 14.1      |
> > |              | Entropy (×1)             | 69.7      | 77.0      | 28.2         | 32.2         | 23.0      | 12.9      |
> > |              | Entropy (×3)             | 71.1      | 80.1      | 22.6         | 29.8         | 15.1      | 12.7      |
> > |              | PC-Sampler (×1)          | 70.1      | 77.3      | 26.6         | 32.2         | 25.4      | 19.1      |
> > |              | PC-Sampler (×3)          | 69.9      | 79.5      | 22.6         | 34.4         | 19.8      | 14.5      |
> > |              | ReMDM                    | 70.4      | 80.1      | 27.4         | 34.0         | 23.4      | 19.9      |
> > |              | **LookUM**               | **74.5**  | **82.3**  | **29.2**     | **35.8**     | **27.3**  | 17.9      |
> >
> >
> >
> > The experimental results clearly demonstrate that **simply increasing the number of paths is not an effective strategy**, and that LookUM remains competitive even when compared under the same inference budget.  In particular, simply increasing the number of paths in the baselines does not reliably improve performance. For difficult tasks(MATH), adding more paths often degrades the results, and even for easier tasks(GSM8K), the performance gain remains marginal.
> >
> > This confirms that LookUM does not rely on brute-force compute scaling; rather, it provides a more effective way to utilize test-time compute. The performance gains of LookUM arise **not from simply spending more compute**, but from its ability to **efficiently avoid local errors by leveraging the certainty score**.
> >
> > The updated experimental results have been added to Table 2 of the revised manuscript, and the discussion regarding LookUM’s computational cost has been included in  L432–L439.

---

> > > ### Author Response · Authors · 2025-11-27
> > >
> > > >W3. The approach performs multiple forward passes for each inference step. Although conceptually simple, this can substantially increase inference time. However, it might not be necessary to perform LookUM during each of the inference steps. It would be useful to explore whether using any method could decide when to apply lookahead unmasking — potentially avoiding unnecessary lookahead steps and reducing the inference cost without significant performance loss.
> > >
> > >
> > > We sincerely appreciate the reviewer’s constructive suggestion aimed at improving our method. Following the reviewer’s guidance, we conducted additional experiments by dividing the generation process into four stages. For convenience, we also summarize the key findings below.
> > >
> > > |  |  ~0.25T |0.25T~ 0.5T | 0.5T~0.75T |0.75T~ T |
> > > | -------- | -------- | -------- | -------- | - |
> > > | GSM8K     | 69.8     |   70.0   | 70.2 | 69.7 |
> > > | MATH500     | 27.8     |   27.0   | 27.8 | 27.6 |
> > > | Countdown     | 25.0     |   22.7   | 20.1 | 21.5 |
> > >
> > > The experimental results show that performance improves, yet remains below that of the original LookUM. This pattern suggests that LookUM’s gains stem from its ability to correct local errors across the entire unmasking trajectory.
> > >
> > > The experimental results and accompanying discussion have been added to Section 4.3 and are included in the revised version at lines L442–L458. We sincerely appreciate the reviewer’s suggestion, which allowed us to better analyze and clarify the operational behavior of LookUM.
> > >
> > >
> > > We sincerely appreciate the valuable suggestions that helped improve our work, and we hope that our response effectively addresses the reviewer’s concerns.
> > >
> > > >Q1. Would LookUM perform better if the verifier did not average negative entropy over all tokens? Perhaps focusing on a subset of the most uncertain tokens could reduce noise in the uncertainty estimation.
> > >
> > > Thank you for this excellent question. We also explored the direction suggested by the reviewer during our internal investigations in hopes of further improving performance. We computed the score in LLaDA using the top 5% most uncertain tokens across the entire sequence and applied LookUM based on this score. Below, we report the corresponding experimental results.
> > >
> > >
> > > |  | GSM8K | MATH500 | Countdown |
> > > | -------- | -------- | -------- | -------- |
> > > | LookUM     | 72.7     |   28.8    | 25.4|
> > > | Uncertain tokens | 70.7     |   28.4   | 23.4 |
> > >
> > >
> > > The experiments show that the performance does not improve meaningfully. To better understand this outcome, we analyzed the token-wise entropy at each step and sorted the all tokens in sequence by their entropy values. We found that many high-entropy tokens exhibit very similar entropy levels, and the set of such tokens changes across timesteps. This variability makes it difficult to reliably identify a consistent subset of uncertain positions to target, which we believe explains the lack of performance gains.
> > >
> > >
> > >
> > >
> > >
> > > >Q2. Is there any intuitive reason of the performance drop as we increase the number of paths (e.g., in MATH500)? How frequently does it occur? One possible explanation is that the estimated uncertainty does not perfectly correlate with the true path quality.
> > >
> > > Thank you for pointing out this important phenomenon. We increased the number of candidate paths and tracked which paths were selected during generation, leading to two key observations:
> > >
> > > * We found cases where the certainty score is high despite the final answer being incorrect. These cases are closely tied to early-termination tokens such as “answer” or “eot”. Because these tokens often exhibit artificially low uncertainty, the model prematurely terminates the sequence before completing the necessary reasoning, resulting in an incorrect final output.
> > >
> > > * In SMC-based approaches, multiple particles are sampled in parallel, naturally expanding the exploration space. As the number of paths increases, the search covers a broader set of trajectories, which raises the likelihood of encountering early-termination tokens. Once such a token appears in any trajectory, degeneration occurs in the following step, where all particles collapse into that early-termination token, ultimately producing an incorrect output.
> > >
> > > This behavior can also be understood as a form of reward hacking, analogous to what has been previously observed in reward-guided generation.
> > >
> > >
> > > We hope that our response has fully addressed the reviewer’s concerns, and we would like to once again express our sincere gratitude for the constructive and thoughtful feedback.

---

### Author Response · Authors · 2025-11-27
**Response to all reviewers**

We sincerely thank all reviewers for their constructive and insightful feedback. LookUM is a simple yet effective test-time decoding method for diffusion LLMs that reduces local unmasking errors through uncertainty-guided 1-step lookahead, and it demonstrates strong and consistent empirical performance across diverse benchmarks. Reviewers highlighted the paper’s consistent empirical strength (`FvKa, Labt, nMFR`), clear presentation (`6sEz,Labt,nMFR`), and thorough ablation studies as key strengths(`6sEz,Labt,nMFR`).

Below, we summarize the main improvements made in response to the reviewers’ comments.

* Compute-matched comparisons
Since LookUM uses up to 3 paths, we re-evaluated all baselines under the same 3-path compute budget. We found that simply increasing the number of paths rarely improves baseline performance and often degrades it, whereas LookUM remains consistently superior under equal compute. We additionally report wall-clock time measurements, showing that LookUM benefits from GPU parallelism and maintains efficient inference time in practice.

* Clarified comparison to related work
We strengthened both quantitative and qualitative comparisons with ReMDM, Train for the Worst, P2, and other related approaches, and clarified how LookUM is structurally distinct from remasking- or planning-based decoding methods.

* Expanded ablation and diagnostic studies
Per reviewer request, we conducted additional experiments on gradient norms, attention entropy, lookahead depth, subset-based entropy, and timestep-specific application. These analyses allow a deeper understanding of LookUM’s mechanisms, robustness, and extensibility.

We carefully incorporated all reviewer suggestions and added every requested experiment, which significantly improved the clarity and completeness of the paper. We sincerely appreciate the reviewers’ valuable feedback.

---

### Meta-Review · Area_Chair_fUZn · 2026-01-06

**Summary:**

Overall, reviewers agreed that the proposed method is conceptually clean and empirically solid, and several found the results consistently positive across tasks. However, the recommendation ultimately hinged on three main considerations. First, reviewers placed significant weight on fairness under matched inference compute and real inference cost, and whether the reported gains should be interpreted as genuine improvements in decoding effectiveness rather than a consequence of increased test-time computation. Second, reviewers emphasized the need for a clear and convincing comparison to closely related work, in particular P2 and other planning or remasking-based decoding approaches, given the similarity in problem formulation and perspective. Third, reviewers highlighted the importance of clarity and completeness in key technical details, including the construction of the lookahead state, the role and interpretation of the verifier, hyperparameter choices, and consistency in experimental reporting. These factors together shaped the overall assessment and informed the final recommendation.

**Reviewer Concerns:**

Addressed concerns: The rebuttal substantially clarified the computational overhead of the proposed method, including explicit accounting of model calls, compute-matched evaluations, and wall-clock time measurements. Several previously unclear technical details were also clarified, including aspects of the lookahead mechanism, verifier behavior, and reporting inconsistencies, which helped resolve concrete questions raised during the initial review phase.

Remaining concerns: Despite these improvements, some reviewers remained unconvinced about the level of novelty relative to existing work, particularly planning- or path-based unmasking methods such as P2. While distinctions were clarified, the contribution was still viewed by part of the reviewer pool as incremental within a closely related line of work. In addition, although compute-matched results were provided, the overall cost–gain tradeoff was still perceived by some as modest, leaving open questions about whether the empirical improvements are sufficiently compelling given the additional inference complexity.

**Reviewer Scores:**

FvKa: likely stays at 6, as the main concerns around compute accounting and practical inference cost were addressed, but broader reservations about impact and tradeoffs remain.

6sEz: likely stays at 4, given continued concerns about the magnitude of gains relative to computational cost and the overall strength of the contribution.

Labt: likely stays at 4, as issues regarding comparison to closely related baselines and perceived incremental novelty were only partially alleviated.

nMFR: likely at 6, reflecting that most substantive technical and comparison-related concerns were addressed during the rebuttal and discussion.

---

### Decision · Program_Chairs · 2026-01-26

Reject